# The genetic legacy of the expansion of Bantu-speaking peoples in Africa

Cesar A. Fortes-Lima[1,28], Concetta Burgarella[1,2,28], Rickard Hammarén[1,28], Anders Eriksson[3], Mário Vicente[4,5], Cecile Jolly[1], Armando Semo[6,7,8], Hilde Gunnink[9,10], Sara Pacchiarotti[9], Leon Mundeke[11], Igor Matonda[11], Joseph Koni Muluwa[12], Peter Coutros[9], Terry S. Nyambe[13], Justin Cirhuza Cikomola[14], Vinet Coetzee[15], Minique de Castro[16], Peter Ebbesen[17], Joris Delanghe[18], Mark Stoneking[19,20], Lawrence Barham[21], Marlize Lombard[22], Anja Meyer[23], Maryna Steyn[23], Helena Malmström[1,22], Jorge Rocha[6,7,8], Himla Soodyall[24,25], Brigitte Pakendorf[26], Koen Bostoen[9] & Carina M. Schlebusch[1,22,27]✉

The expansion of people speaking Bantu languages is the most dramatic demographic event in Late Holocene Africa and fundamentally reshaped the linguistic, cultural and biological landscape of the continent[1–7]. With a comprehensive genomic dataset, including newly generated data of modern-day and ancient DNA from previously unsampled regions in Africa, we contribute insights into this expansion that started 6,000–4,000 years ago in western Africa. We genotyped 1,763 participants, including 1,526 Bantu speakers from 147 populations across 14 African countries, and generated whole-genome sequences from 12 Late Iron Age individuals[8]. We show that genetic diversity amongst Bantu-speaking populations declines with distance from western Africa, with current-day Zambia and the Democratic Republic of Congo as possible crossroads of interaction. Using spatially explicit methods[9] and correlating genetic, linguistic and geographical data, we provide cross-disciplinary support for a serial-founder migration model. We further show that Bantu speakers received significant gene flow from local groups in regions they expanded into. Our genetic dataset provides an exhaustive modern-day African comparative dataset for ancient DNA studies[10] and will be important to a wide range of disciplines from science and humanities, as well as to the medical sector studying human genetic variation and health in African and African-descendant populations.

African populations speaking Bantu languages (Bantu-speaking populations (BSP)) constitute about 30% of Africa's total population, of which about 350 million people across 9 million km[2] speak more than 500 Bantu languages[1,11]. Archaeological, linguistic, historical and anthropological sources attest to the complex history of the expansion of BSP across subequatorial Africa, which fundamentally reshaped the linguistic, cultural and biological landscape of the continent. There is a broad interdisciplinary consensus that the initial spread of Bantu languages was a demic expansion[2,4–7,12–15] and ancestral BSP migrated first through the Congo rainforest and later to the savannas further east and south[2,4,5,7,15–23]. However, debates persist on the pathways and modes of the expansion (Supplementary Fig. 1 and Supplementary Note 1) (see refs. 18,19 for linguistic-based syntheses).

Whereas most recent human expansions involved latitudinal movements through regions with similar climatic conditions[24,25], the expansion of the BSP is notable for its primarily longitudinal trajectory, traversing regions with highly diverse climates and biomes, including the highlands of Cameroon, central African rainforests, African savannas and arid south-western Africa. Despite consensus on its demic nature, genetic studies of the BSP expansion have not revealed the

[1]Human Evolution Program, Department of Organismal Biology, Uppsala University, Uppsala, Sweden. [2]AGAP Institut, University of Montpellier, CIRAD, INRAE, Institut Agro, Montpellier, France. [3]cGEM, Institute of Genomics, University of Tartu, Tartu, Estonia. [4]Centre for Palaeogenetics, University of Stockholm, Stockholm, Sweden. [5]Department of Archaeology and Classical Studies, Stockholm University, Stockholm, Sweden. [6]CIBIO, Centro de Investigação em Biodiversidade e Recursos Genéticos, Universidade do Porto, Vairão, Portugal. [7]BIOPOLIS Program in Genomics, Biodiversity and Land Planning, CIBIO, Campus de Vairão, Vairão, Portugal. [8]Departamento de Biologia, Faculdade de Ciências, Universidade do Porto, Porto, Portugal. [9]UGent Centre for Bantu Studies (BantUGent), Department of Languages and Cultures, Ghent University, Ghent, Belgium. [10]Leiden University Centre for Linguistics, Leiden, the Netherlands. [11]University of Kinshasa, Kinshasa, Democratic Republic of Congo. [12]Institut Supérieur Pédagogique de Kikwit, Kikwit, Democratic Republic of Congo. [13]Livingstone Museum, Livingstone, Zambia. [14]Faculty of Medicine, Catholic University of Bukavu, Bukavu, Democratic Republic of Congo. [15]Department of Biochemistry, Genetics and Microbiology, University of Pretoria, Pretoria, South Africa. [16]Biotechnology Platform, Agricultural Research Council, Onderstepoort, Pretoria, South Africa. [17]Department of Health Science and Technology, University of Aalborg, Aalborg, Denmark. [18]Department of Diagnostic Sciences, Ghent University, Ghent, Belgium. [19]Department of Evolutionary Genetics, Max Planck Institute for Evolutionary Anthropology, Leipzig, Germany. [20]Laboratoire de Biométrie et Biologie Evolutive, UMR 5558, Université Lyon 1, CNRS, Villeurbanne, France. [21]Department of Archaeology, Classics & Egyptology, University of Liverpool, Liverpool, UK. [22]Palaeo-Research Institute, University of Johannesburg, Johannesburg, South Africa. [23]Human Variation and Identification Research Unit, School of Anatomical Sciences, Faculty of Health Sciences, University of the Witwatersrand, Johannesburg, South Africa. [24]Division of Human Genetics, School of Pathology, Faculty of Health Sciences, University of the Witwatersrand, Johannesburg, South Africa. [25]Academy of Science of South Africa, Pretoria, South Africa. [26]Dynamique Du Langage, UMR5596, CNRS & Université de Lyon, Lyon, France. [27]SciLifeLab, Uppsala, Sweden. [28]These authors contributed equally: Cesar A. Fortes-Lima, Concetta Burgarella, Rickard Hammarén. ✉e-mail: carina.schlebusch@ebc.uu.se

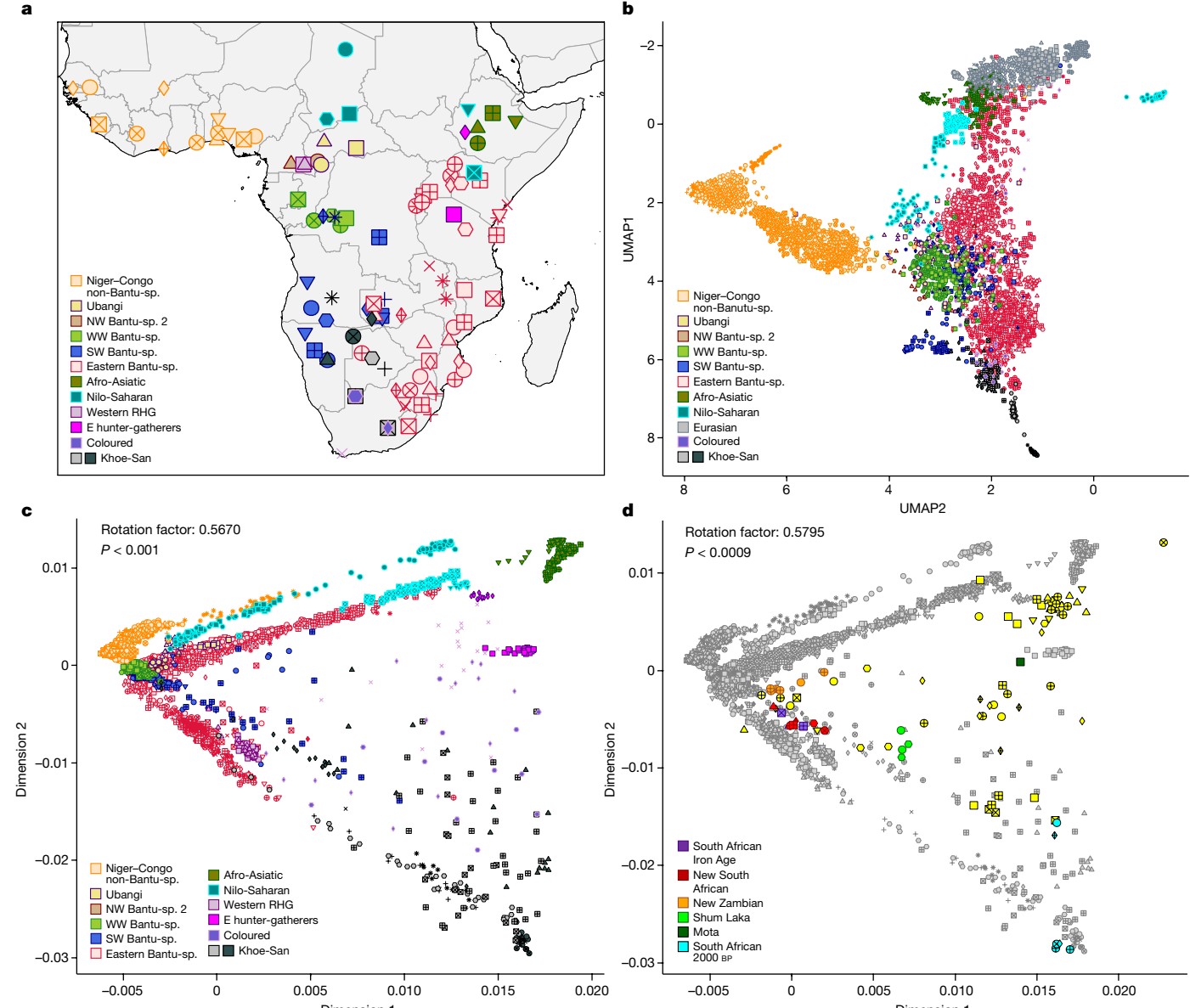

**Fig. 1 | Population structure within sub-Saharan African populations.**
**a**, Geographical locations of the 111 sub-Saharan African populations selected for population genetic analysis within the AfricanNeo dataset. Populations with the same colour belong to the same group based on linguistic and geographic characterization (Supplementary Table 2). **b**, UMAP analysis of selected populations included in the AfricanNeo dataset. **c**, Procrustes rotated PCA of sub-Saharan African populations included in the Only-African dataset

(Procrustes correlation to geography < 0.567, *P* < 0.001). **d**, Procrustes rotated PCA for projected aDNA individuals (with colours; Supplementary Fig. 95) and present-day sub-Saharan African populations (in grey, same as **c**) included in the Africa-aDNA dataset (Procrustes correlation > 0.580, *P* < 0.0009). Symbol shapes are defined in Supplementary Figs. 3, 4. Bantu-sp., Bantu-speaking. Vector basemap and map tiles were provided by CartoDB under a Creative Commons licence CC BY 4.0 (2023).

typical serial-founder effect observed when small migrant groups settle in new areas, leading to a decrease in genetic diversity with increasing distance from the putative homeland[13,26,27]. This might be a result of subsequent genetic diversity increases from admixture with local populations or long-distance interactions with later Bantu-speaking migrations, known as 'spread-over-spread' events[2,20,28,29]. This underlying complexity, coupled with the different migration histories proposed by linguistics, archaeology and genetics, makes the expansion of BSP interesting for exploration with newer population genetic methods and modelling approaches that are spatiotemporally sensitive.

Although whole-genome studies of African populations have become available recently[30–33] and some localized genome-wide genotype studies exist[4,14,15], comprehensive genomic data for BSP across sub-Saharan Africa remain limited. To better understand their large-scale spread,

we collected a genome-wide genotype dataset of 1,763 individuals (Supplementary Table 1), including 1,526 Bantu-speaking individuals from 147 BSP across 14 African countries and 237 other sub-Saharan African individuals (Supplementary Fig. 2 and Extended Data Fig. 1). This dataset includes 117 populations not represented in previous genomic studies and covers most main branches of the Bantu language family[18]: north-western 2 (2 NW-BSP 2), west-western (7 WW-BSP), south-western (13 SW-BSP plus the Damara, a Khoe-Kwadi speaking population from Namibia sharing a genetic profile with BSP) and eastern (44 E-BSP) (Fig. 1a and Supplementary Table 2). In addition, we generated genomic data for 12 ancient individuals from Late Iron Age sites of south-central and southern Africa (present-day Zambia and South Africa), spanning 97–688 years before present (BP). This comprehensive dataset allowed us to explore the demographic history of BSP using

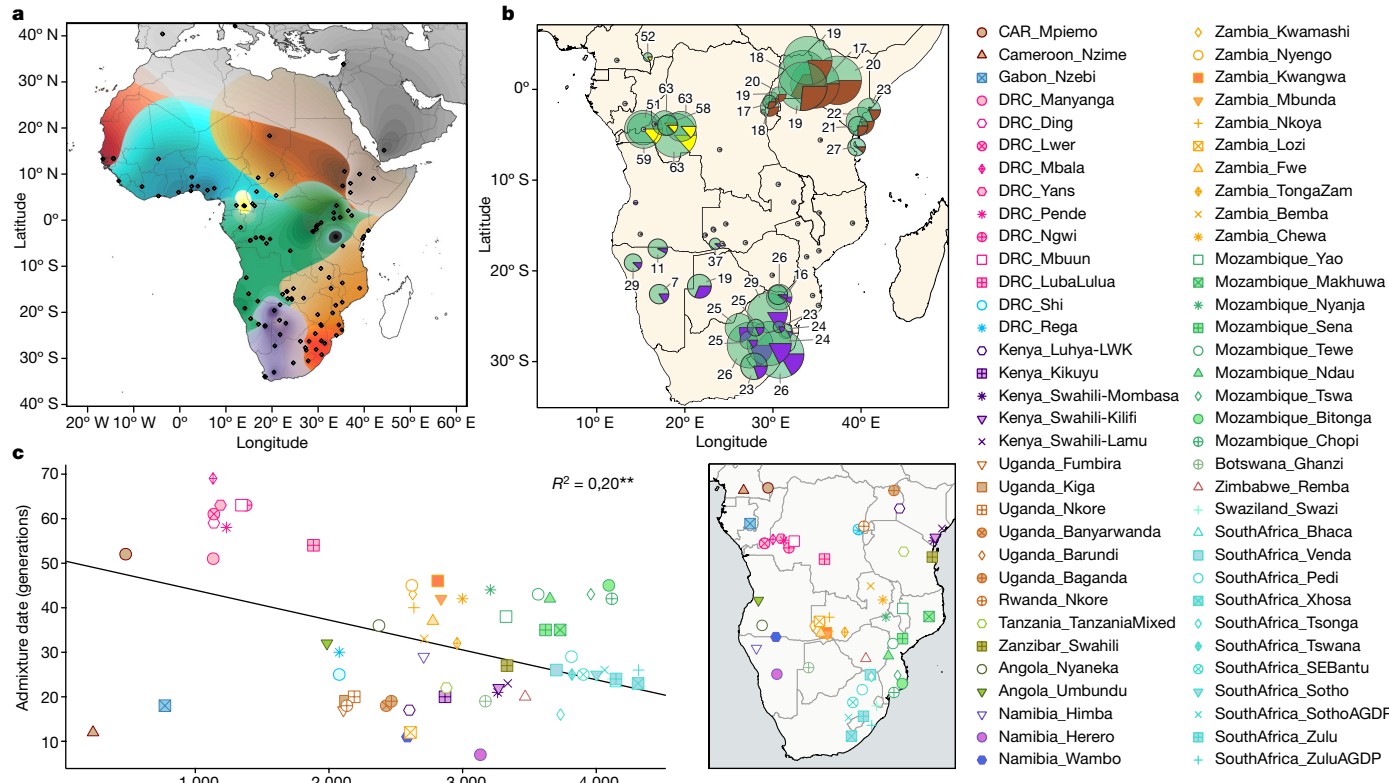

**Fig. 2 | Population structure, admixture dates and fractions. a**, Contour map of overlapping unsupervised ADMIXTURE results at K = 12 created using the Kriging method for all the populations included in the AfricanNeo dataset. Ancestry components with values under 25% are not represented on the map (all the values estimated for each ancestry component are shown in Extended Data Fig. 3). **b**, Inferred admixture dates (number of generations ago) and fractions (pie chart) for each BSP estimated using MOSAIC analyses. Each inferred population source is highlighted with a different colour: WCA-related ancestry in green; western RHG ancestry in yellow; Afro-Asiatic-speaking ancestry in brown; and Khoe-San ancestry in purple. The size of the charts is in relation to the sample size of each population. **c**, Linear regression of admixture dates of studied BSP versus geographical distances from Cameroon (coloured according to country of origin). Vector basemap and map tiles were provided by CartoDB under a Creative Commons licence CC BY 4.0 (2023).

allele-frequency and haplotype-based methods, genetic diversity summary statistics and spatial modelling providing insights into African human history.

## Admixture influences genetic structure

After quality control and merging with publicly available data from representative ethnolinguistic groups (Supplementary Information), we assembled a dataset of 4,950 individuals from 124 populations (111 sub-Saharan African and 13 Eurasian populations with at least 10 individuals per population), hereafter referred to as the 'AfricanNeo' dataset (Supplementary Figs. 3–5 and Supplementary Table 2). To visually represent genetic affinities between populations, we applied four dimensionality reduction methods (Supplementary Information) and found evidence for fine-scale population structure between sub-Saharan African populations with a clear geographical component and a broad correspondence with the main linguistic groups in Africa (Fig. 1b,c, Extended Data Fig. 2 and Supplementary Figs. 6–13). Population substructure within BSP (NW-BSP 2, WW-BSP, SW-BSP and E-BSP) can be distinguished (Supplementary Note 2).

Population substructure and suggestions of admixture are also apparent in model-based clustering analyses (Fig. 2a, Supplementary Figs. 14–27 and Supplementary Note 2) and show a finer representation of population ancestries with three main BSP-associated genetic components (Extended Data Figs. 3 and 4 and Supplementary Table 3): the dark-green component found in most BSP, the light-blue component shared between non-Bantu Niger-Congo and western BSP (NW-BSP 2,

WW-BSP and SW-BSP) and the orange component mainly found in south-eastern BSP.

BSP also show differential genetic affinities with other populations (Fig. 1b,c, Extended Data Fig. 2 and Supplementary Table 4). This pattern may be the result of genetic admixture with local populations, during and/or after the expansion of BSP across subequatorial Africa[7,13,34–36]. We have formally tested the hypothesis of admixture and its regional character using f3 and f4 statistics (Supplementary Information). The results confirm significant and differential contributions of Afro-Asiatic-related ancestry in eastern BSP from Kenya and Uganda, of western rainforest hunter-gatherer (wRHG)-related ancestry in western BSP from the Democratic Republic of Congo (DRC) and the Central African Republic (CAR), and of Khoe-San-related ancestry in southern BSP from South Africa, Botswana, Zambia (Fwe population) and Namibia (Extended Data Table 1, Supplementary Figs. 28–32 and Supplementary Tables 5 and 6). These findings underscore the intricate genetic history of BSP, characterized by distinct admixture patterns with diverse local groups in specific geographic regions of subequatorial Africa (Supplementary Note 2).

## BSP-specific population substructure

To assess if admixture with local groups is the main process driving spatial patterns of substructure in BSP (Figs. 1c and 2a), we masked out admixed genomic regions in BSP[37] keeping only west-central African (WCA) genomic components (Supplementary Information). This masked dataset allowed us to minimize the influence of

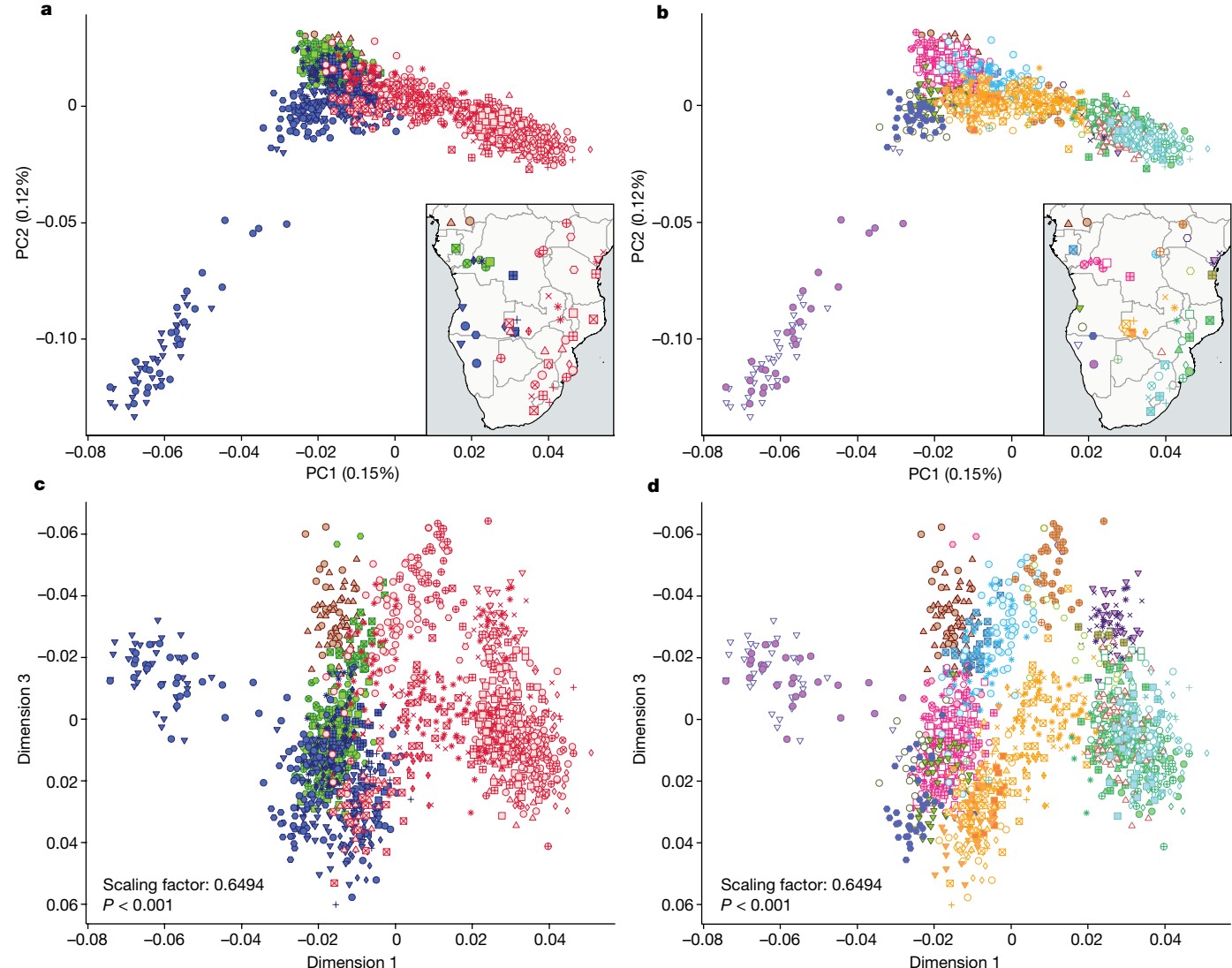

**Fig. 3 | Population structure patterns in BSP after admixture masking.**
**a**–**d**, PCA plots on the admixture-masked BSP dataset show BSP coloured by linguistic groups in PC1 versus PC2 (**a**) and PC1 versus PC3 (**c**), and by geography represented by countries in PC1 versus PC2 (**b**) and PC1 versus PC3 (**d**). Procrustes rotation was used for **c** and **d** and the estimated scaling factor was 0.649 (*P* < 0.001). Further details and legends are included in Supplementary Fig. 34. Vector basemap and map tiles were provided by CartoDB under a Creative Commons licence CC BY 4.0 (2023).

non-Bantu-speaker ancestries in subsequent analyses (Supplementary Fig. 33). Principal component analysis (PCA) on the admixture-masked dataset (Supplementary Note 3) shows that BSP retain a clear genetic structure which aligns with geographic (Fig. 3b,d) and linguistic (Fig. 3a,c) features, suggesting that processes other than genetic admixture influence spatial patterns of BSP diversity. However, this structure could also be driven by outlier BSP with increased genetic drift, for example, Herero and Himba from Namibia largely influence PC2 (Figs. 3a,b and Supplementary Fig. 34). We therefore repeated the PCA after excluding Himba and Herero from the analysis and still observed population substructure in the remaining BSP (Extended Data Fig. 5).

We then investigated whether BSP harbour signatures of genetic isolation and population size changes potentially driving the observed structure. Most BSP show similar patterns of genetic drift reflected in runs of homozygosity (ROH) (Supplementary Figs. 35–47, Supplementary Table 7 and Supplementary Note 4) and changes in the effective population sizes ($N_e$) that support population expansion signatures in the past 10–30 generations (Supplementary Figs. 49–51 and Supplementary Note 5). The Himba and Herero populations notably deviate from the general patterns of BSP, showing higher values of the genomic inbreeding coefficient (Supplementary Figs. 39 and 48), higher averages for the longest ROH length categories (Supplementary Table 7 and Supplementary Fig. 41) and higher intensities of founder events ($I_f$ = 1.6% and 1.2%, respectively; Supplementary Figs. 52 and 53 and Supplementary Table 8) than do other BSP (Supplementary Notes 4 and 5). These signatures can be the consequence of genetic isolation since their arrival in south-western Africa or shared cultural practices, for example, endogamic practices linked to cattle herding lifeways[38], as suggested by mitochondrial DNA data[39], genotype data[40] and exome sequencing data[38]. The early twentieth century Herero genocide by imperial Germany is not expected to trigger an increased ROH signal, as this was a relatively recent event in the context of population demographic histories (Supplementary Note 4).

## Models underlying BSP substructure

Exploring different models and analysing their fit to the observed genetic data can yield valuable insights into the population history that underlies the genetic patterns and geographic distribution of

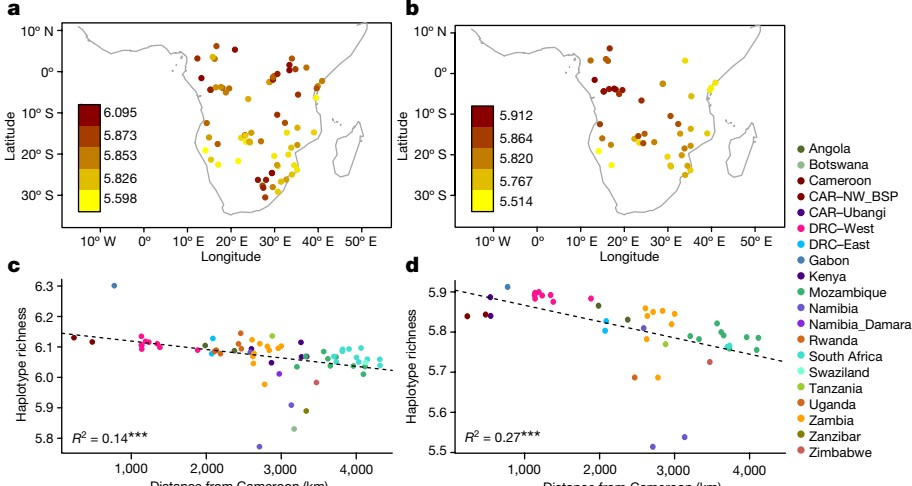

**Fig. 4 | Patterns of genetic diversity in BSP. a**, Map of haplotype richness estimated for the unmasked AfricanNeo dataset showing only BSP (*n* = 67 populations; with a minimum sample size of 10 individuals and a maximum size of 30 individuals). **b**, Haplotype richness estimated for the admixture-masked BSP dataset (*n* = 49 populations) that includes BSP with at least 70% of WCA-related ancestry. **c**,**d**, Decrease in haplotype richness estimates with geographical distance from Cameroon to the sampling location of each BSP included in the unmasked BSP dataset of **a** (**c**) and in the admixture-masked BSP dataset of **b** (**d**). Dashed lines represent the linear regression between haplotype richness estimates and geographical distances.

BSP. A strong correlation between genetic relatedness and geography suggests an isolation-by-distance (IBD) model, which assumes stepwise gene flow between neighbouring groups leading to gradients of genetic affinity across geographic space. Our dataset of BSP fits an IBD pattern (Supplementary Figs. 54–57 and Supplementary Information), including when admixture is removed (Supplementary Figs. 58–61 and Supplementary Note 6), consistent with previous findings based on fewer BSP and a smaller dataset[2]. Alternative models could, however, explain these patterns. For instance, under a serial-founder model we also expect a strong correlation between shared genetic ancestry and geography. However, in contrast to IBD models, a serial-founder model would also show a decrease in genetic diversity from the putative region of origin. To distinguish between these two models, we investigated the spatial distribution of three genetic diversity summary statistics (haplotype richness, haplotype heterozygosity and linkage disequilibrium; Supplementary Information) suitable for array-based genotype data[12]. The estimated statistics support a serial-founder model in which the highest genetic diversity is found in western BSP with a steady decline with distance towards eastern and southern BSP (Supplementary Figs. 62–65 and Supplementary Note 7). This pattern is stronger in the admixture-masked dataset (Fig. 4, Extended Data Fig. 6 and Supplementary Figs. 66–75). Further evidence supports serial-founder dynamics during the BSP expansion from west-central Africa; for example, significant demographic founder events have been inferred in 19 BSP (Supplementary Table 8 and Supplementary Note 5) and a maximum-likelihood tree of the admixture-masked BSP dataset shows north-western BSP 2 at the base of the tree and most eastern BSP forming a monophyletic group (Extended Data Fig. 7, Supplementary Fig. 76 and Supplementary Note 8). By contrast, admixture largely drove the shape of the maximum-likelihood trees for the unmasked datasets (Supplementary Figs. 77–80).

Overall, these analyses support the suggestion that the expansion of BSP started from west-central Africa and spread mostly through serial bottlenecks throughout subequatorial Africa. The negative correlation between genetic diversity and distance from the source, even in the unmasked dataset, suggests admixture had a small impact on genetic diversity of BSP, either because there was not that much gene flow with indigenous groups or some BSP moved on before they received substantial local gene flow. The fact that the admixture patterns are largely region-specific (Fig. 2a and Extended Data Fig. 3)—that is, in each population, we detect non-BSP ancestry from local groups and not from elsewhere—suggests the latter.

## Routes and timing of the BSP expansion

To gain a deeper insight into how the expansion of BSP unfolded, we investigated the spatial routes and timing of their movements. First, we used a climate-informed spatially explicit model[9,41] to infer the most likely initial expansion routes, with specific scenarios of population expansion that correspond to the 'late-split' and 'early-split' hypotheses proposed by linguistic studies (Supplementary Fig. 81). We ran one million Wright–Fisher simulations to test three expansion scenarios that differ in whether BSP were allowed to spread south through the Congo rainforest (that is, 'southern route' or late-split; Supplementary Fig. 1b), north of the rainforest ('northern route' or early-split; Supplementary Fig. 1a) or both routes. In each simulation, gene genealogies were generated for selected African populations[30] (Supplementary Table 9). The scenario with only the northern route received substantially less statistical support from the data compared to scenarios for both routes or only the southern route ($r^2$ = 0.19, 0.32 and 0.34, respectively; Supplementary Information). Therefore, results support the late-split hypothesis, in agreement with recent linguistic, archaeological and genetic evidence[4,15,18–20,42] and highlight the importance of the Congo rainforest in the initial expansion of BSP.

Previous studies proposed gene flow between the western and eastern branches of Bantu speakers[2,5]. Even though populations speaking western and eastern Bantu languages are more separated in the PCA towards the terminal parts of the distribution, there is overlap toward the middle, particularly in BSP from current-day Zambia and the DRC (Fig. 3). These two countries thus represent interaction zones between different linguistic subgroupings, which is also reflected in their genetic composition (Supplementary Figs. 26 and 27). Our inference of BSP expansion routes by tracing nearest genetic distance (fixation index ($F_{ST}$)) values over the geographic landscape also indicates Zambia as a possible interaction nexus (Fig. 5a, Supplementary Figs. 82 and 83 and Supplementary Note 9). Specifically, the Lozi population represents the

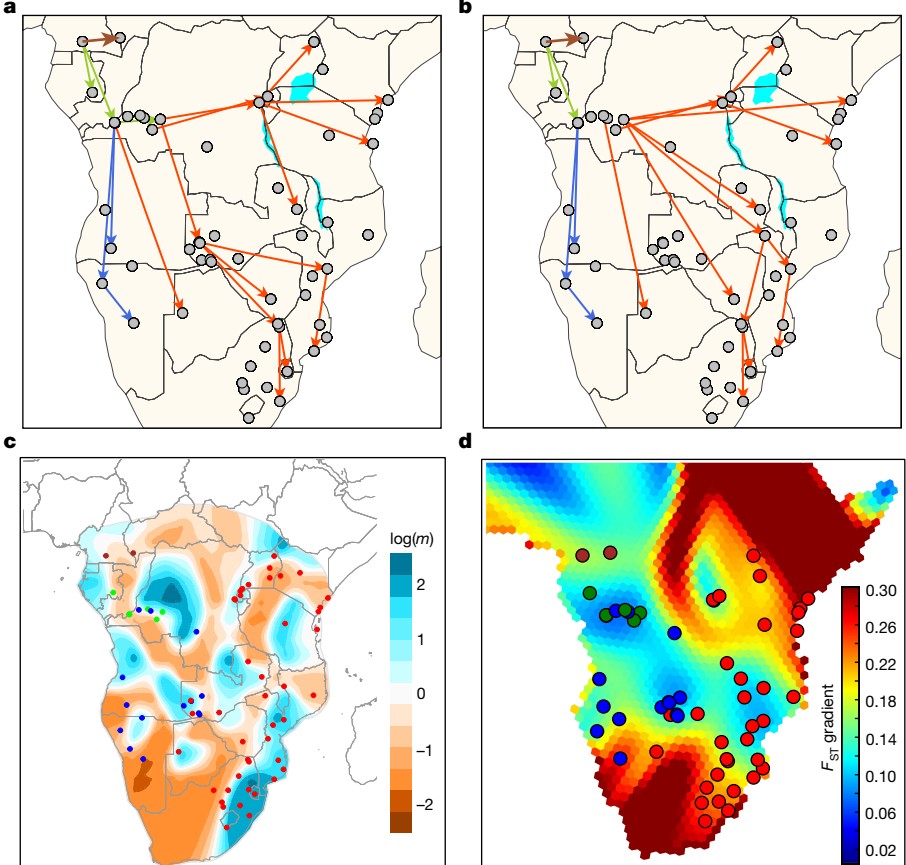

**Fig. 5 | Migration routes and rates in BSP. a,b,** Putative migration routes of BSP inferred using pairwise $F_{ST}$ values (**a**) and after removing the Zambian Lozi population from the analyses (**b**). Arrow colours correspond to north-western Bantu speakers 2 (NW-BSP 2; brown; one arrow between Cameroon and CAR), west-western Bantu speakers (WW-BSP; green), south-western Bantu speakers (SW-BSP; dark blue) and eastern Bantu speakers (E-BSP; red). **c,** Spatial visualization of effective migration rates (EEMS software) estimated with the masked Only-BSP dataset. log($m$) denotes the effective migration rate on a $\log_{10}$ scale, relative to the overall migration rate across the habitat. Populations are coloured according to each Bantu-speaking linguistic group (brown, green, dark blue and red dots). **d,** GenGrad analysis using $F_{ST}$ as the genetic distance for the admixture-masked BSP dataset. Hexagons of the grid were plotted with a colour scale representing the $F_{ST}$ gradient (key).

proxy population of the Bantu-speaking migrants from the western DRC to Zimbabwe, Mozambique, Eswatini (former Swaziland) and South Africa (Supplementary Fig. 83a,c). However, the Lozi language, widely used as a *lingua franca* in Zambia's Western Province and adjacent areas, was only introduced into the region in the nineteenth century CE by Sotho-speaking immigrants from what is today South Africa[43,44]. Removing the Lozi population from the analysis moves the connection point between eastern and south-eastern BSP with western BSP to the western DRC (Fig. 5b and Supplementary Fig. 83b).

Current-day Zambia as a point of divergence between expansion routes of BSP was previously proposed by ref. 30 using a few BSP not representative of the whole Bantu-speaking area. Here, with notably better geographic representation of BSP, we identified Zambia and the western DRC as important nexus zones. However, spatially explicit analyses using EEMS, FEEMS, MAPS and $F_{ST}$ estimates (Supplementary Information, Fig. 5c,d and Supplementary Figs. 84–91) and clustering methods (Extended Data Fig. 4) suggest barriers to gene flow and population structure even in these zones (Supplementary Note 10), possibly caused by the linguistic division between BSP. Future research and model-testing methods could further establish whether these are interaction zones between populations speaking the eastern and south-western branches of Bantu languages or splitting points in past expansion routes. Spatial methods further indicate high effective migration rates along the Indian Ocean coast from Kenya to eastern South Africa (blue areas in Fig. 5c and Supplementary Fig. 87), as was

reported previously[15], together with longitudinal zones of lower migration in the central parts of the continent (brown and dark red areas in Fig. 5c,d, respectively).

Dating of admixture events[45] strongly supports the main direction of BSP expansion across subequatorial Africa (Supplementary Note 11). Admixture dates (Fig. 2b) significantly correlate with geographic distance from the BSP homeland ($r^2 = 0.20$, $P = 2.6 \times 10^{-5}$; Fig. 2c), with earlier admixture dates in west-central Africa and more recent dates toward the extremes of the expansion (Supplementary Figs. 92 and 93 and Supplementary Table 10). These results also suggest that the rate of movement of BSP was more or less constant through time, despite the wide variety of environments and population interactions. Admixture dates seem to be older than expected in BSP from western regions (for example, between BSP and wRHG in western DRC) and younger in certain eastern regions (for example, between BSP and different eastern African groups in Uganda and Kenya) (Fig. 2b,c), suggesting that the rates of movement of BSP into these regions were either faster or slower than the average speed or that the admixture occurred earlier or later after arrival than in the other regions. Further investigation into the sociocultural aspects of the interactions with the linguistically and culturally diverse populations and the environmental challenges encountered by BSP during their expansion, particularly in adapting to diverse ecological zones and acquiring new subsistence practices, presents a promising avenue for future cross-disciplinary research.

## Spread-over-spread events versus continuity

The initial BSP expansion across subequatorial Africa may have been followed by subsequent migrations along similar routes, creating a pattern of spread-over-spread events[46]. In some cases, these later migrations might have replaced earlier settlers and their languages[20,29,47]. Consequently, certain branches of the Bantu language family tree may no longer accurately represent the initial BSP expansion[3]. This raises questions about the reliability of using only lexical and geographical data from modern Bantu languages for phylogeographic analyses to depict the ancestral BSP migration[18,19]. Contact and admixture between incoming and previously settled Bantu-speaking groups could lead to genetic data reflecting a mixture of migration events, whereas linguistic data may represent only the latest spread event. As a result, both linguistics and genetics may correlate with geography but not necessarily with each other.

We tested this using Mantel tests (Supplementary Tables 11 and 12). Pairwise-population linguistic and geographic distances are significantly correlated ($r$-statistics = 0.6457; $P$ = 0.0002), as are genetic and geographic distances ($r$-statistics = 0.1666; $P$ = 0.0158). However, the correlation between genetic and linguistic distances is not significant ($r$-statistics = 0.0104; $P$ = 0.4153). The correlation between genetics and geography increases after controlling for linguistic data, as well as between linguistics and geography after controlling for genetics (Supplementary Information). A marginally significant negative correlation between linguistic and genetic data is observed after controlling for geography ($r$-statistics = −0.1291; $P$ = 0.0496). This overall weaker correlation between genetics and linguistics (whereas both correlate strongly with geography) could point to separate histories underlying the genetic and linguistic data that could involve secondary, and potentially more localized, spread waves. Other explanations are also possible, for example, admixture between linguistically distantly related BSP.

To further explore the possibility of spread-over-spread events, we compared the genetic diversity of present-day BSP and ancient (aDNA) individuals from Africa, including whole-genome sequencing data of 12 individuals from this study (97–688 years BP) and data from 83 individuals (150–8,895 years BP) from previous aDNA studies (Extended Data Fig. 1 and Supplementary Table 13). See refs. 8,48 and Supplementary Table 14 for the archaeological and morphological descriptions and dating of the sequenced individuals (Supplementary Fig. 94 and Supplementary Information). Dimensionality reduction and clustering analyses represent genetic affinities between aDNA and modern-day individuals (Fig. 1d, Extended Data Fig. 8, Supplementary Figs. 95–98 and Supplementary Note 12). In South Africa, Late Iron Age aDNA individuals (since 688 BP) show homogeneity and genetic affinity with local modern BSP (Extended Data Table 2, Supplementary Figs. 99–101 and Supplementary Table 15), thus largely supporting a scenario of genetic continuity since the Late Iron Age. Our new Late Iron Age aDNA individuals from Zambia (since 311 BP), however, have a more heterogeneous genetic makeup showing genetic affinities with modern BSP from a wider geographical area (Supplementary Figs. 98 and 102–104). This supports the suggestion that Zambia might have been a crossroad for different movements of BSP.

## New and comprehensive genomic dataset

Our dataset demonstrates its potential to provide an effective modern-day background genetic dataset to compare with aDNA individuals (Fig. 1d, Extended Data Fig. 8 and Supplementary Note 12). The underlying historical patterns in BSP are very difficult to distinguish on the basis of modern-day data only. Both IBD and serial-founder models can represent more complex underlying population histories among studied BSP, such as multiple overlapping expansions from the same location following similar routes. A clear manifestation of this pattern

has been seen in the comparison of European history inferences based on modern DNA[49] and aDNA[50]. Analyses such as our Mantel test correlations between linguistics, geography and genetics tentatively point to complex histories and possible spread-over-spread events (Supplementary Table 11), in agreement with recent archaeological studies[20] (Supplementary Note 1). Future aDNA studies on human remains from different archaeological contexts, associated with the Early, Middle and Late Iron Age in Africa, as well as different pottery traditions, will be necessary for assessing the affinity of the Bantu-speaker-related ancestry to each other and to current-day BSP. Therefore, the availability of our extensive genomic dataset, encompassing the full geographic expansion range of BSP, will enable further testing of these spread-over-spread proposals using aDNA.

## Conclusion

Our study supports a large demic expansion of BSP with ancestry from western Africa spreading through the Congo rainforest to eastern and southern Africa in a serial-founder fashion. This finding is supported by patterns of decreasing genetic diversity and increasing $F_{ST}$ from their point of origin, as well as admixture dates with local groups that decrease with distance from western Africa. Although our genetic findings provide less precision compared to existing linguistic models[18,19], they caution against relying solely on modern language data for tracing BSP dispersal because of potential spread-over-spread events and genetic admixture between linguistically distantly related BSP. Our genetic findings highlight the need for a comprehensive interdisciplinary study into how the demographic history of BSP influenced their language evolution. The significant correlation of admixture times with distance from the BSP source argues for a relatively constant rate of BSP expansion despite the extremely heterogeneous nature of the landscape. Although there were corridors of higher and lower effective migration rates across the African landscape, current-day Zambia and the DRC seem to be important crossroads or interaction points for the expansion of BSP. Future aDNA studies using our dataset as comparative data and new spatial modelling methods will refine our understanding of BSP expansion and their interactions with other African populations. The new findings and data will be useful not only to population geneticists, archaeologists, historical linguists, anthropologists and historians focusing on population history in Africa but also to the medical and health sector studying human genetic variation and human health in African and African-descendant populations.

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

## Methods

### Genotyping and assembled datasets

In total, 1,763 samples encompassing 163 African populations were collected with informed consent in 14 sub-Saharan African countries (Supplementary Figs. 2–4 and Supplementary Table 1). Ethical permits and sampling permission were obtained in African countries and the study as a whole was approved by the Swedish Ethical Review Board (DNR-2021-01448). DNA samples were genotyped at the SNP&SEQ Technology Platform, NGI/SciLifeLab Genomics (Sweden). We used seven genotyping batches on the Illumina Infinium H3Africa Consortium array (about 2.4 million single nucleotide polymorphisms (SNPs)) and one batch on the Illumina HumanOmni2.5-Octo BeadChip. After merging all newly genotyped data and quality control steps using PLINK v.1.90b6.4 (ref. 51), genotype data consisted of 2,221,827 autosomal SNPs (Supplementary Fig. 5). After removing 67 samples because of low genotyping rate and 105 individuals because of their first- or second-degree kinship with other samples, we obtained 1,591 individuals and 2,221,827 SNPs for the 'genotyped' dataset. After merging the genotyped dataset with comparative data and performing quality control steps, we assembled the 'Full-Genotyped' dataset that contains 482,459 SNPs and 5,341 individuals from 227 populations (including 81 populations with sample sizes lower than 10 individuals) and three subdatasets with selected African populations. We included 4,950 individuals from 124 African and Eurasian populations in the AfricanNeo dataset (Supplementary Fig. 3a,b); 3,902 individuals from 111 sub-Saharan African populations in the 'Only-African' dataset (Supplementary Fig. 4a,c); and 2,108 individuals from 67 populations speaking Bantu languages (BSP) in the 'Only-BSP' dataset (Supplementary Fig. 4b,d). BSP with fewer than ten individuals were removed from specific analyses. To avoid sample-size biases, for some analyses (for example, local ancestry inference and analyses using the masked and imputed data) populations with large sample sizes were randomly downsampled to 30 individuals and we obtained 1,495 individuals from 124 populations in the downsampled admixture-masked Only-BSP dataset.

### Dimensionality reduction and clustering methods

To visualize genetic variation and population structure in BSP, we applied four dimensionality reduction methods for genome-wide SNP data. We first used the uniform manifold approximation and projection (UMAP) approach[52] directly on the genotype data. Second, we applied PCA using smartpca[53]. To combine the information of the first ten principal components, we then used the PCA-UMAP approach[54]. Fourth, we used the genotype convolutional autoencoder (GCAE) method[55]. In addition, we applied an unsupervised clustering-based approach using ADMIXTURE software v.1.3.0 (ref. 56) and cluster numbers ranging from K = 2 to K = 25.

### Ancient DNA samples

To compare the genetic affinities of ancient and present-day BSP, we merged the AfricanNeo dataset with 12 aDNA individuals from southern and south-central Africa (at present Zambia and South Africa) and 83 aDNA individuals from previous studies[23,57–62] (Extended Data Fig. 1, Supplementary Fig. 95 and Supplementary Table 13). We then projected the aDNA individuals onto a background of present-day populations using PCA. After merging haplodized modern samples and pseudohaplodized aDNA individuals and performing quality control and linkage disequilibrium-pruning steps, we used unsupervised ADMIXTURE analysis from K = 2 to K = 12.

### Runs of homozygosity

We used PLINK to calculate five parameters of ROH in BSP and worldwide populations: mean ROH size, total length of ROH, sum of short ROH, sum of long ROH and ROH-based inbreeding coefficient (or $F_{ROH}$).

For each studied population, we also calculated six ROH length classes. To estimate effective population sizes over the past 50 generations, we used IBDNe[63]. To infer both the age and strength of demographic founder events in BSP, we used ASCEND v.10 (ref. 64). To identify significant founder events, we followed the four criteria recommended by ref. 64.

### Admixture timing analysis and admixture masking

To estimate admixture dates, we applied haplotype-based admixture inference methods. We used MOSAIC v.1.4 (ref. 45) with two- and three-way admixture models for each BSP included in the AfricanNeo dataset. For haplotype phasing of the AfricanNeo dataset, we used SHAPEIT v.2.r904 (ref. 65). For local ancestry inference, we estimated haplotypic admixture from six reference panels in individuals from BSP using RFMix software v.1.5.4 (ref. 37). To avoid the influence of admixture patterns in BSP in our ancestry-specific analyses, we removed haplotypes with non-WCA-related ancestry from each haploid genome of each Bantu-speaking individual using a masking approach (Supplementary Figs. 105–107). For each assembled dataset, we explored patterns of population structure between and within populations using smartPCA[53].

### Phylogenetic analyses and correlations

To investigate phylogenetic relationships between all the BSP, we used TreeMix v.1.13 (ref. 66). The likelihood of each proposed population-based maximum-likelihood TreeMix topology was assessed by bootstrapping blocks of 500 SNPs and assigning Ju/'hoansi from Namibia as the root of the population tree. To test the correlation between genetic, linguistic and geographical distances, we performed Mantel tests and partial Mantel tests using the R package ncf[67]. As genetic distances, we computed pairwise $F_{ST}$ between populations included in the ancestry-masked Only-BSP dataset using EIGENSOFT package v.6 (ref. 68). Geographic distances were calculated as pairwise great circle distances between the studied populations using the R package geosphere[69]. For linguistic distances, we used a linguistic dataset from the multistate matrix of cognate sets identified by ref. 18. In total, 38 BSP matched the genetic dataset and the linguistic dataset of 409 Bantu languages studied by ref. 18.

### Patterns of genetic diversity

To investigate spatial patterns of genetic diversity of studied African populations, we calculated statistics based on haplotype diversity and linkage disequilibrium information. Haplotype heterozygosity and haplotype richness were computed following recommendations from ref. 12 with homemade scripts implemented in Python. Each calculation was also repeated ten times. Values were calculated per chromosome and then averaged across the genome of each individual in each population. We characterized linkage disequilibrium patterns in each population with more than ten individuals by measuring the correlation coefficient ($r^2$) between all pairs of SNPs within 500 kilobase pairs windows using PLINK. To assess whether haplotype heterozygosity, haplotype richness and linkage disequilibrium patterns in BSP were consistent with a history of expansion from the homeland of BSP, we performed a linear regression between the three summary statistics and geographical distances from Cameroon, assuming that the BSP expansion started in that region[3].

### Pairwise genetic distances

To reconstruct potential routes of expansion of BSP, pairwise $F_{ST}$ values were calculated between one population from Cameroon (Nzime) and each of the studied BSP for the masked and imputed Only-BSP dataset. We also applied the GenGrad method from ref. 70 but using $F_{ST}$ as the genetic distance metric and with slightly adjusted parameters to better fit the smaller study area.

## Effective migration rates

To further investigate spatial population structure in sub-Saharan African populations, we used EEMS software[71] and its implementation FEEMS[72] and MAPS[73]. EEMS and FEEMS were performed on the Only-African and Only-BSP datasets before and after using the masking approach and MAPS was performed on the masked Only-BSP dataset. EEMS analysis was repeated three times and an average was taken as input for the visualization as recommended in the EEMS manual.

## Testing isolation-by-distance models

To test four models of migration, we used SpaceMix v.0.13 (ref. 74). The software generates geogenetic maps in which genetics rather than physical distances determine the distances between individuals/populations. The general underlying assumption evaluated with SpaceMix is that under an IBD pattern, geographic and geogenetic positions will be similar, which is a pattern of IBD. The best-fitting model was evaluated using Pearson correlations between the expected and observed data.

## Testing models of migration routes

To test different demographic scenarios for the BSP expansion, we used a spatiotemporally explicit population genetic framework[9,41]. Here, we adapted the extension of the model presented by ref. 41 to apply multiple local expansions for different scenarios of expansion. We considered three demographic scenarios in which the expansion of BSP proceeded north of the rainforest, south through a rainforest corridor or using both northern and southern routes. For each demographic scenario, we ran one million simulations with parameters drawn from an independent uniform distribution for parameters characterizing the BSP expansion and from a log-uniform distribution for parameters describing the initial global expansion of anatomically modern humans taken from ref. 41.

## Ethics and inclusion

This study was conducted according to the Declaration of Helsinki (World Medical Association, 2013). DNA samples were collected with informed consent from participants. Ethical permits and sampling permission were obtained in African countries (Methods) and the study as a whole was approved by the Swedish Ethical Review Board (DNR-2021-01448). The sampling for this study emerged from population genetic and archaeology projects that involved local research institutions and the participation of local communities. Local institution involvement included research design, selection of archaeological material for analyses, modern-day DNA sample collection, community involvement, permit clearance, feedback on analyses, help with writing and feedback on the manuscript. Before submission of this study for publication, the corresponding author and first author participated in-person in the ICHG conference held in Cape Town, as well as held online presentations and meetings with local researchers.

## Reporting summary

Further information on research design is available in the Nature Portfolio Reporting Summary linked to this article.

## Data availability

SNP array genotype data of modern-day African populations and whole-genome data of aDNA individuals generated in this project were made available through the European Genome-Phenome Archive (EGA) data repository (EGA accessory nos. EGAS50000000006 and EGAS00001007519 for modern and aDNA, respectively). Controlled-access policies guided by participant consent agreements will be implemented by the AfricanNeo Data Access Committee (AfricanNeo DAC accessory no. EGAC00001003398). Authorized NIH DAC granted data access to C.M.S. for the controlled-access genetic data deposited in the NIH dbGAP repository (accession code phs001396. v1.p1 and project ID 19895). C.M.S. was granted data access to whole-genome sequencing data deposed by the H3Africa Consortium (EGA dataset accessory nos. EGAD00001004220, EGAD00001004316, EGAD00001004334, EGAD00001004393, EGAD00001004448, EGAD00001004505, EGAD00001004533, EGAD00001004557 and EGAD00001005076). Interactive map-based visualizations were created using the Python library bokeh v.3.0.0 and maps were provided by CartoDB (CARTO 2023), other base maps were provided by GoogleMaps (Google 2023) or created using Python libraries (plotly v.5.17.0 and shapely v.1.8.4); R packages (rworldmap v.1.3.6, plotmaps v.1.0, rEEMSplots and rEEMSplots2); and one inhouse vector map in MapInfo interchange format based on the WGS-84 projection.

## Code availability

Code and interactive plots used for plotting are available in two online repositories (GitHub https://github.com/Schlebusch-lab/Expansion_of_BSP_peer-reviewed_article and figshare https://doi.org/10.6084/m9.figshare.24107718).

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

**Acknowledgements** We acknowledge and thank all study participants and local fieldwork teams who helped with the collection of samples. We would like to thank R. P. Stjerna for assistance with sampling for aDNA analyses and G. Mudenda, Director of the Livingstone Museum, for his support and letter of approval to sample the archaeological human remains housed in the Raymond A. Dart collection. We are also indebted to the curators of the Raymond A. Dart Archaeological Human Remains Collection and the University of Pretoria

Bone Collection for permission to study the remains. The remains were sampled and exported under SAHRA permits 2789 and 2835. Isotopic measurements were done at the Uppsala Tandem laboratories. The genotyping and sequencing were performed by the SNP&SEQ Technology Platform, NGI/SciLifeLab Genomics. This facility is part of the National Genomics Infrastructure supported by the Swedish Research Council for Infrastructures and Science for Life Laboratory (NGI-SciLifeLab, Sweden) and is also supported by the Knut and Alice Wallenberg Foundation. The computations/data handling were enabled by resources provided by the National Academic Infrastructure for Supercomputing in Sweden (NAISS) at UPPMAX, partially funded by the Swedish Research Council through grant no. 2022-06725. We thank N. Chousou-Polydouri for technical assistance with the linguistic data matrix. We thank the H3Africa Consortium for sharing the whole-genome sequencing H3Africa data. The views expressed in this paper do not represent the views of either the H3Africa Consortium or their funders, the National Institutes of Health (USA) and Wellcome Trust (United Kingdom). This project was funded by the European Research Council (ERC) under the European Union's Horizon 2020 research and innovation programme: AfricanNeo Project awarded to C.M.S. (grant no. 759933). The aDNA work in the project is funded by the Knut and Alice Wallenberg Foundation grant to C.M.S. Sample collection in the DRC was funded by the ERC-CoG awarded to K.B. for the BantuFirst project (grant no. 724275). Sample collection in Zambia was funded by the Max Planck Society. B.P. acknowledges support from the ASLAN Project of the Université de Lyon (grant no. ANR-10-LABX-0081) in the French programme 'Investments for the Future' operated by the National Research Agency (ANR). C.A.F.-L. received support from the Marcus Borgströms Foundation and the Sven and Lilly Lawski Foundation. C.B. received funding from the European Union's Horizon 2020 research and innovation programme under the Marie Skłodowska-Curie Fellowship Programme (grant no. 839643). H.G. and S.P. received support from the Research Foundation Flanders (postdoctoral grant nos. 12P8423N and 12ZV721N, respectively). H.M. was supported by the Swedish Research Council (grant no. 2017-02503) and by the Riksbankens Jubileumsfond (grant no. P21-0266). A.E. was supported by the European Union's Horizon 2020 research and innovation programme (grant no. 810645) and through the European Regional Development Fund (project no. MOBEC008).

**Author contributions** C.M.S. conceived the study. C.A.F.-L., C.B., R.H., A.E. and M.V. carried out the analyses. M.V., C.J., L.M., I.M., J.K.M., P.C., T.S.N., J.C.C., V.C., M.d.C., P.E., J.D., M. Stoneking, L.B., M.L., A.M., M. Steyn, H.S., B.P., K.B. and C.M.S. contributed to sample collection and preparation. C.A.F.-L., C.B., R.H., A.E., M.V., A.S., H.G., S.P., M. SStoneking, H.M., J.R., B.P., K.B. and C.M.S. contributed to the interpretation of the results. C.A.F.-L., C.B., R.H., B.P. and C.M.S. took the lead in writing the manuscript. All authors provided critical feedback, discussed the results and contributed to the final manuscript. C.M.S. supervised the project.

**Funding** Open access funding provided by Uppsala University.

**Competing interests** The authors declare no competing interests.

### Additional information
**Correspondence and requests for materials** should be addressed to Carina M. Schlebusch.

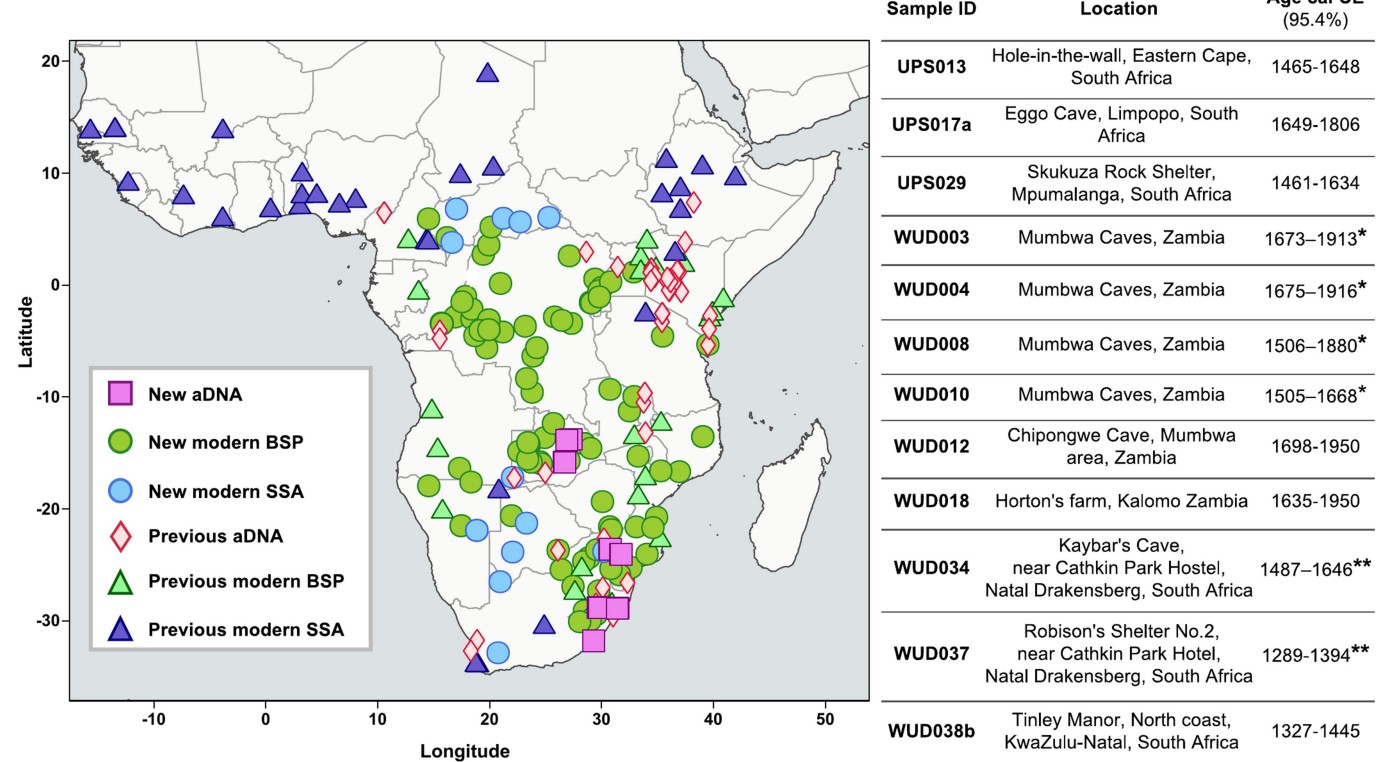

| Sample ID | Location | Age cal CE (95.4%) |
|---|---|---|
| **UPS013** | Hole-in-the-wall, Eastern Cape, South Africa | 1465-1648 |
| **UPS017a** | Eggo Cave, Limpopo, South Africa | 1649-1806 |
| **UPS029** | Skukuza Rock Shelter, Mpumalanga, South Africa | 1461-1634 |
| **WUD003** | Mumbwa Caves, Zambia | 1673–1913* |
| **WUD004** | Mumbwa Caves, Zambia | 1675–1916* |
| **WUD008** | Mumbwa Caves, Zambia | 1506–1880* |
| **WUD010** | Mumbwa Caves, Zambia | 1505–1668* |
| **WUD012** | Chipongwe Cave, Mumbwa area, Zambia | 1698-1950 |
| **WUD018** | Horton's farm, Kalomo Zambia | 1635-1950 |
| **WUD034** | Kaybar's Cave, near Cathkin Park Hostel, Natal Drakensberg, South Africa | 1487–1646** |
| **WUD037** | Robison's Shelter No.2, near Cathkin Park Hotel, Natal Drakensberg, South Africa | 1289-1394** |
| **WUD038b** | Tinley Manor, North coast, KwaZulu-Natal, South Africa | 1327-1445 |

**Extended Data Fig. 1 | Geographical locations of modern and ancient populations.** Geographic information on modern populations and ancient individuals that were included in this study. Figure showing the locations of 12 new aDNA individuals (pink squares) sequenced in this study and new populations genotyped in this study: 147 Bantu-speaking populations (BSP; $n$ = 1,526 individuals; green circles) and 16 sub-Saharan African (SSA) populations that are non-BSP ($n$ = 237; blue circles), together with comparative data from 83 previously published aDNA individuals (red diamonds), 21 BSP ($n$ = 967; green

triangles) and 30 SSA that are non-BSP ($n$ = 1,548; dark blue triangles). Archaeological and biological information about ancient DNA individuals from Zambia and South Africa (SA) presented in this study for the first time is included in the additional table and in Supplementary Table 14. Radiocarbon dates are reported for the first time in this study, except for * from Steyn et al.[8] and ** from Meyer et al.[48] Vector basemap and map tiles were provided by CartoDB (© CARTO 2023).

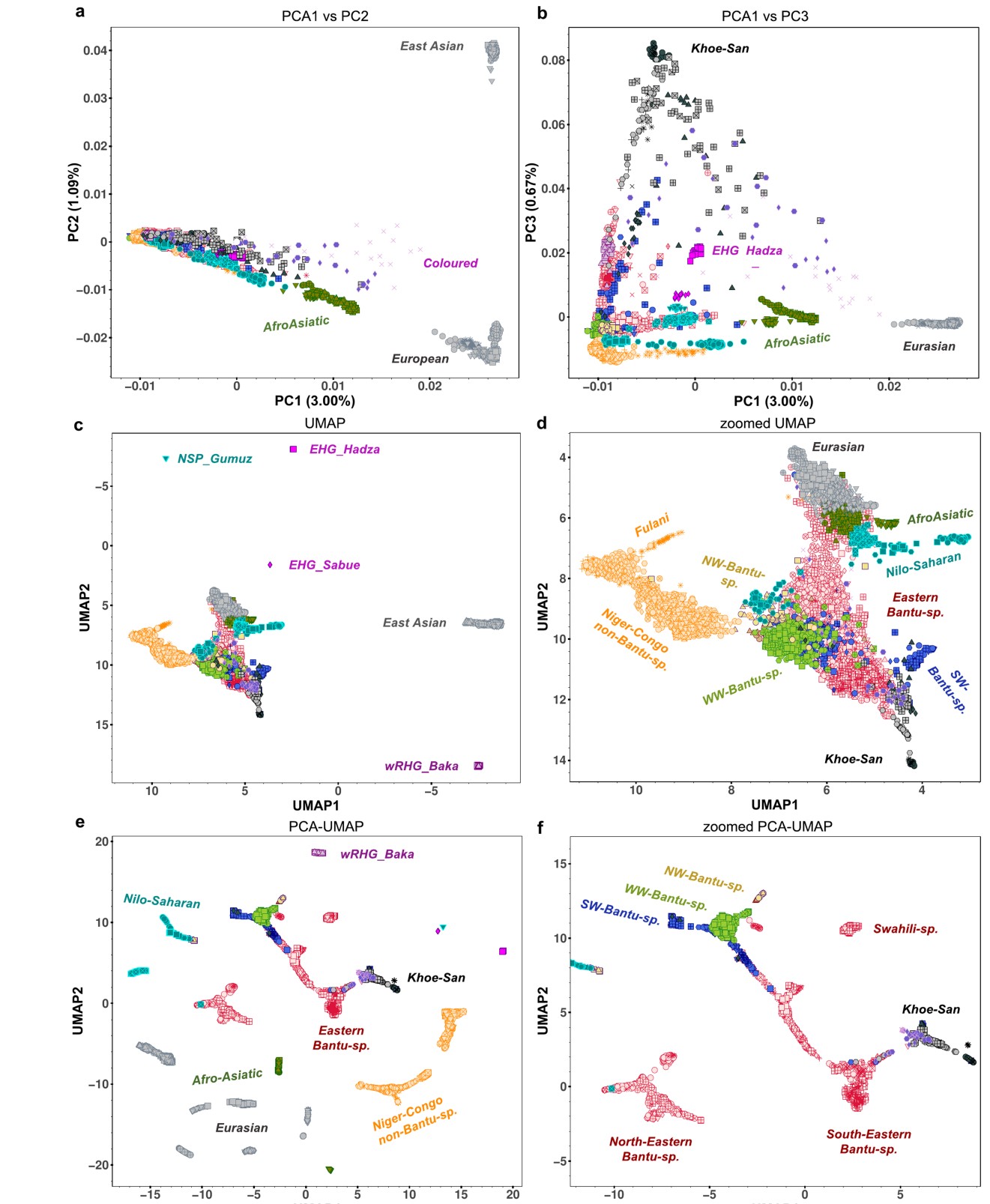

**Extended Data Fig. 2 | Dimensionality reduction methods applied in this study.** Multidimensional spaces including all the populations from the AfricanNeo dataset analysed using **a–b**, PCA; **c–d**, UMAP; and **e–f**, PCA-UMAP. Further details and legends are included in Supplementary Figs. 11a–b, 6 and 12, respectively.

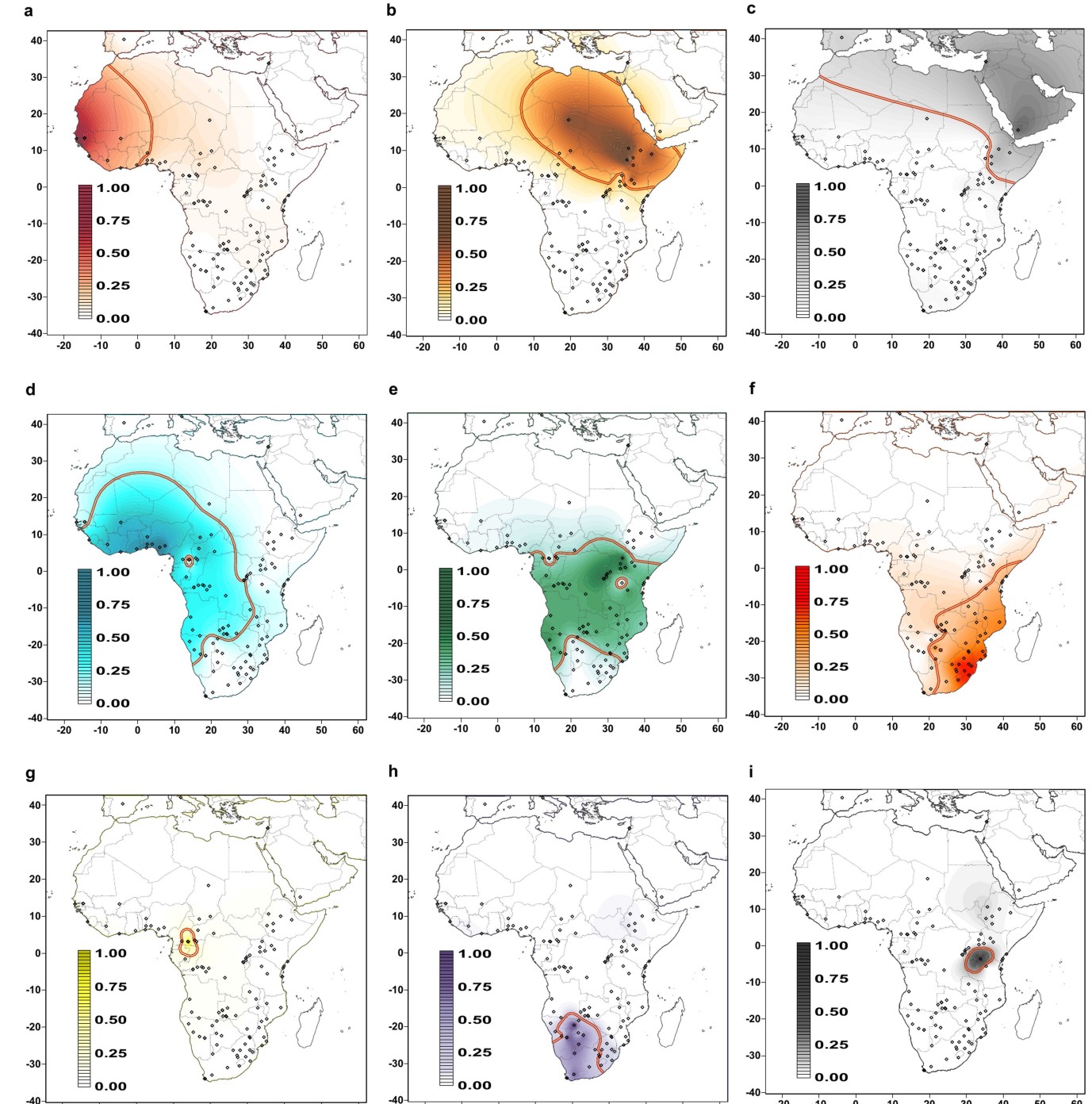

**Extended Data Fig. 3 | Contour maps of each ADMIXTURE result at K = 12.** Panel figure showing contour maps of unsupervised ADMIXTURE at K = 12 were created using the Kriging method for all the populations included in the unmasked AfricanNeo dataset. Average admixture proportions across Africa were depicted for nine main components across the African continent: **a**, red component or west African-related ancestry; **b**, brown component or eastern African-related ancestry; **c**, grey component or Middle Eastern-related ancestry; **d**, blue component or west-central African-related ancestry; **e**, green component or Bantu-speaking-related ancestry; **f**, orange component or south-eastern Bantu-speaking-related ancestry; **g**, yellow component or western RHG-related ancestry; **h**, purple component or Khoe-San-speaking-related ancestry and **i**, black component or eastern RHG-related ancestry. The red line highlights the threshold of 20% used for each component to select the areas plotted together in Fig. 2a.

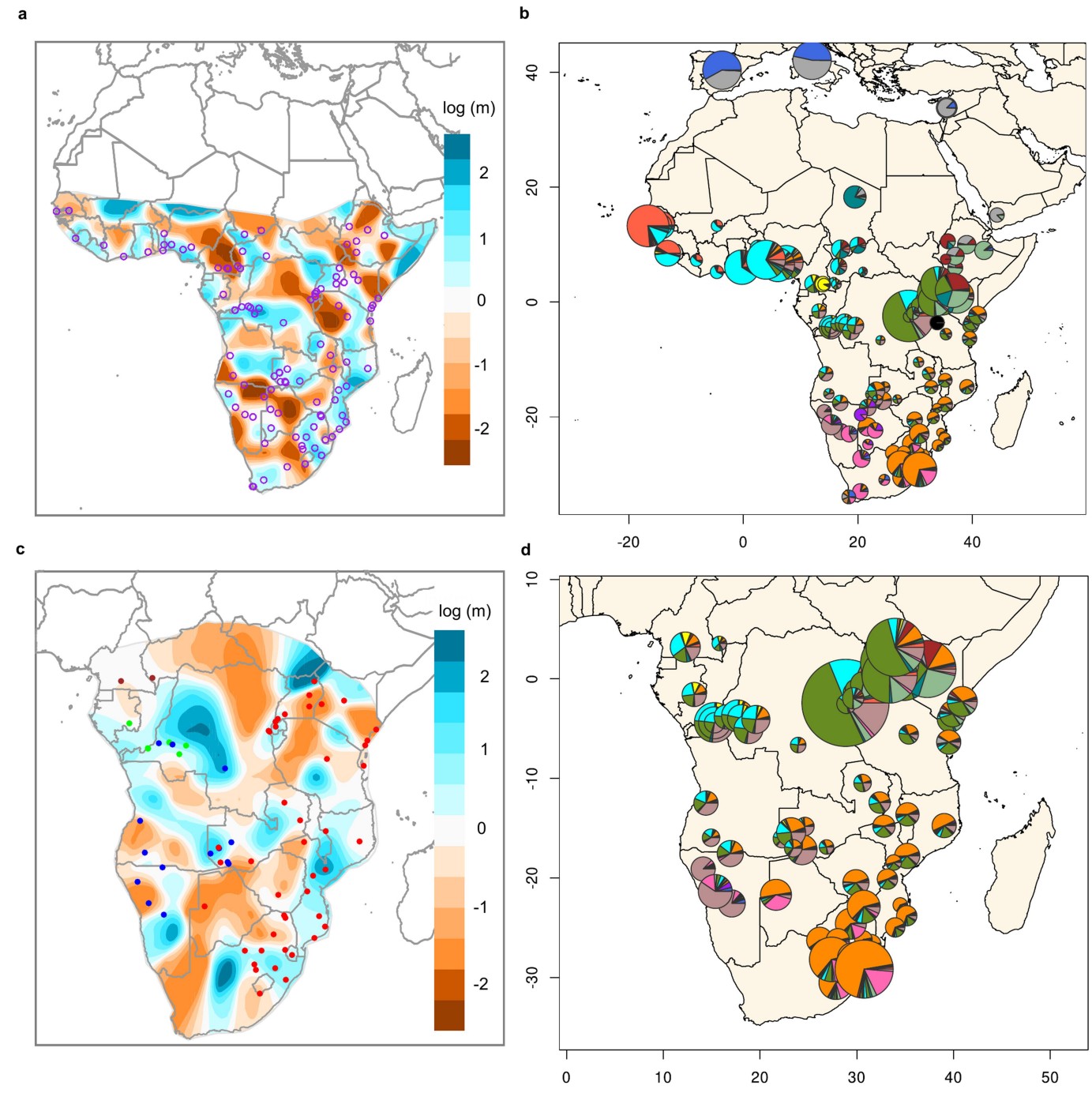

**Extended Data Fig. 4 | Comparisons between EEMS and ADMIXTURE results.** Figure comparing results obtained using EEMS and ADMIXTURE for different unmasked datasets of sub-Saharan African populations or only BSP included in the AfricanNeo dataset: **a**, EEMS results for the AfricanNeo dataset (Supplementary Fig. 84); **b**, ADMIXTURE results at K = 16 for the AfricanNeo dataset (Supplementary Fig. 21); **c**, EEMS results for the Only-BSP dataset (Supplementary Fig. 86); and **d**, ADMIXTURE results at K = 16 showing only the results for BSP included in (b) (Supplementary Fig. 22d).

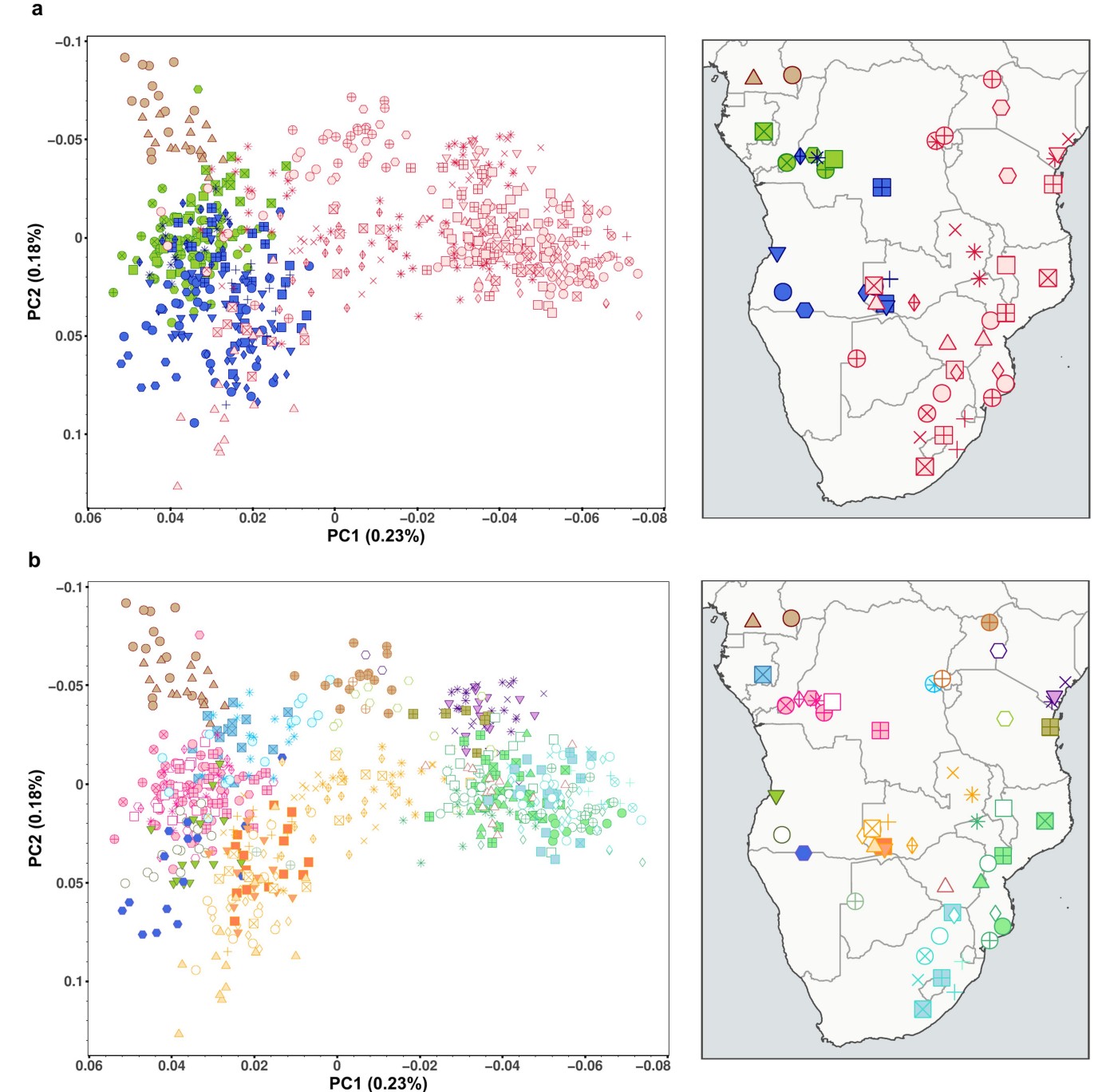

**Extended Data Fig. 5 | Ancestry-specific PCA after removing Himba and Herero populations.** Figure showing PCA plot of BSP **a**, colored by linguistic group and **b**, colored by geography represented by countries, included in the masked Only-BSP dataset after removing Himba and Herero from the analyses. Legend is the same as in Supplementary Fig. 34, except for the Himba and Herero populations that were not included on the maps. Vector basemap and map tiles were provided by CartoDB (© CARTO 2023).

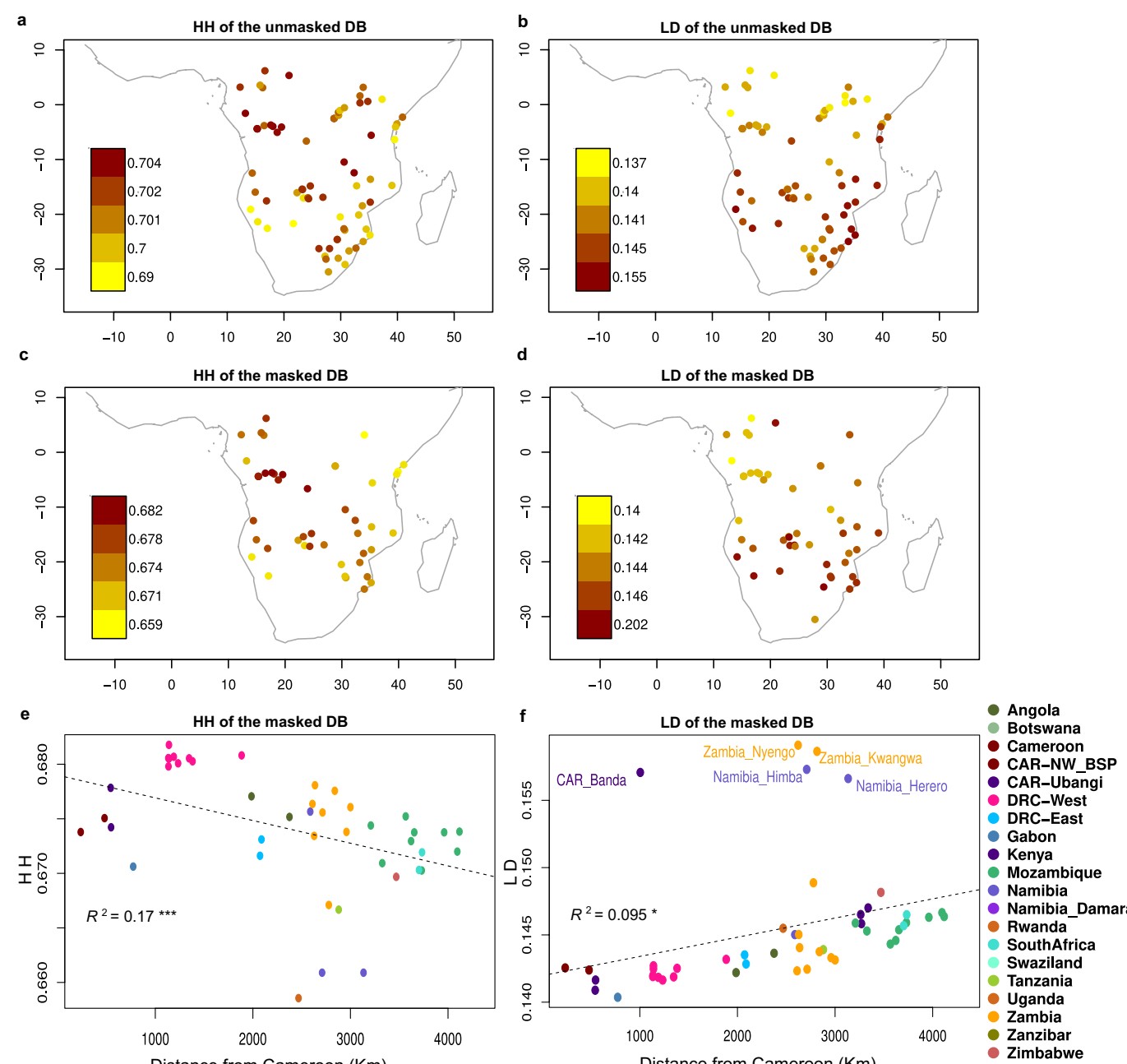

**Extended Data Fig. 6 | Haplotype heterozygosity (HH) and linkage disequilibrium (LD) distribution in BSP.** HH (left column) and LD (right column) estimates were calculated for the unmasked and masked Only-BSP dataset (DB) including only populations with a minimum sample size of 10 individuals and a maximum sample size of 30. HH was estimated on a window size of 50 kb and LD was estimated as $r^2$ at 50 kb (Supplementary Information). Figure showing for each population of the unmasked DB: **a**, HH estimates (based on 67 populations in total); and **b**, LD estimates (based on 70 populations). Figure showing for each population of the masked DB: **c**, HH estimates; **d**, LD estimates; **e**, decrease of HH estimates with geographical distance from Cameroon; and **f**, increase of LD estimates with geographical distance from Cameroon (based on 49 populations; populations with LD > 0.27 were excluded from this analysis). In **e** and **f**, spatial distances were calculated as the spherical distance from each population and a centroid position located in the centre of Cameroon. The dotted line represents the linear regression between the estimates of each statistic and the geographical distances.

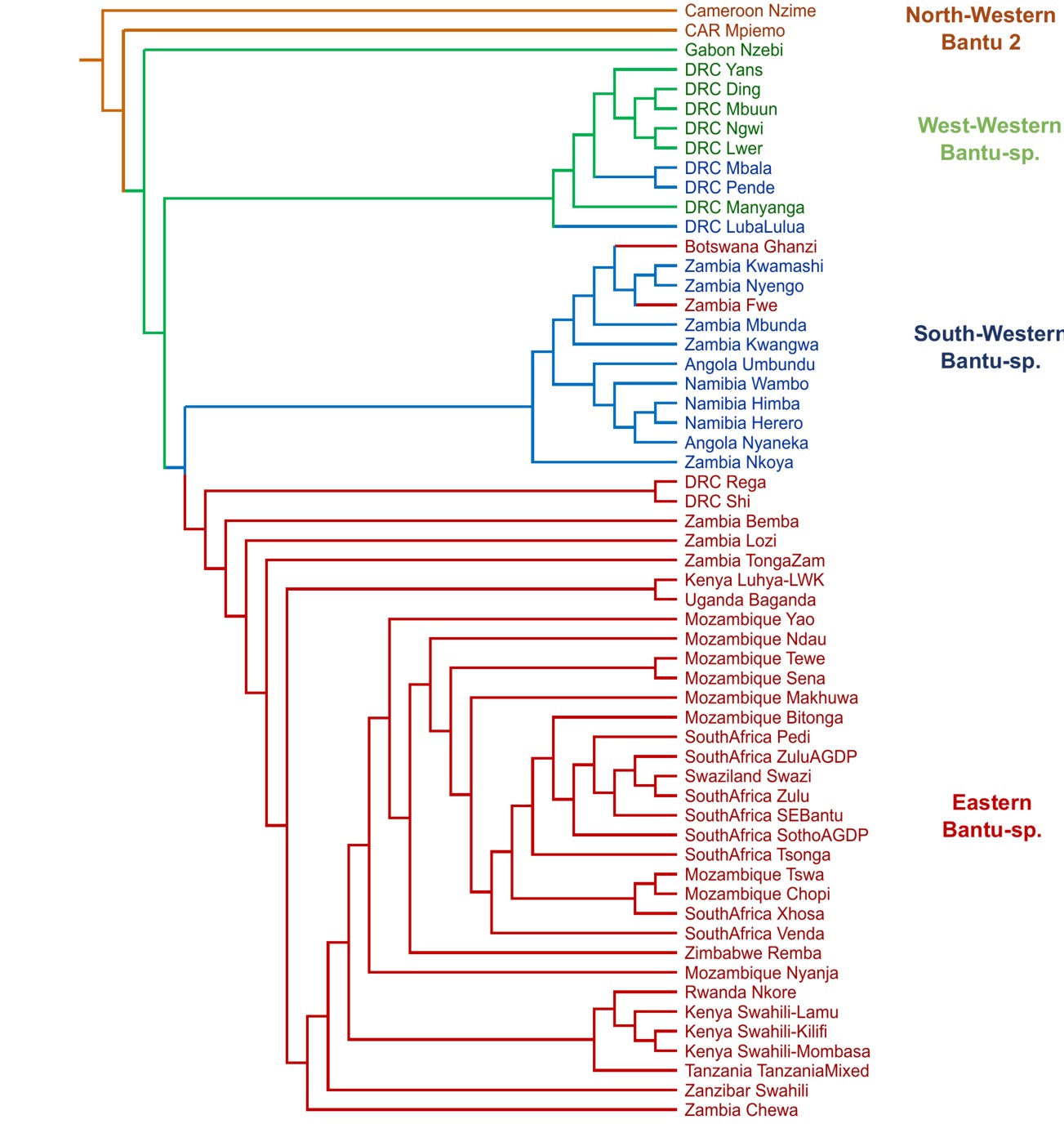

**Extended Data Fig. 7 | TreeMix for the masked Only-BSP dataset.** Figure showing population tree results for the masked Only-BSP dataset in a rectangular shape. Populations are colored by linguistic group. The coancestry matrix of the inferred maximum-likelihood (ML)-tree was included in Supplementary Fig. 76.

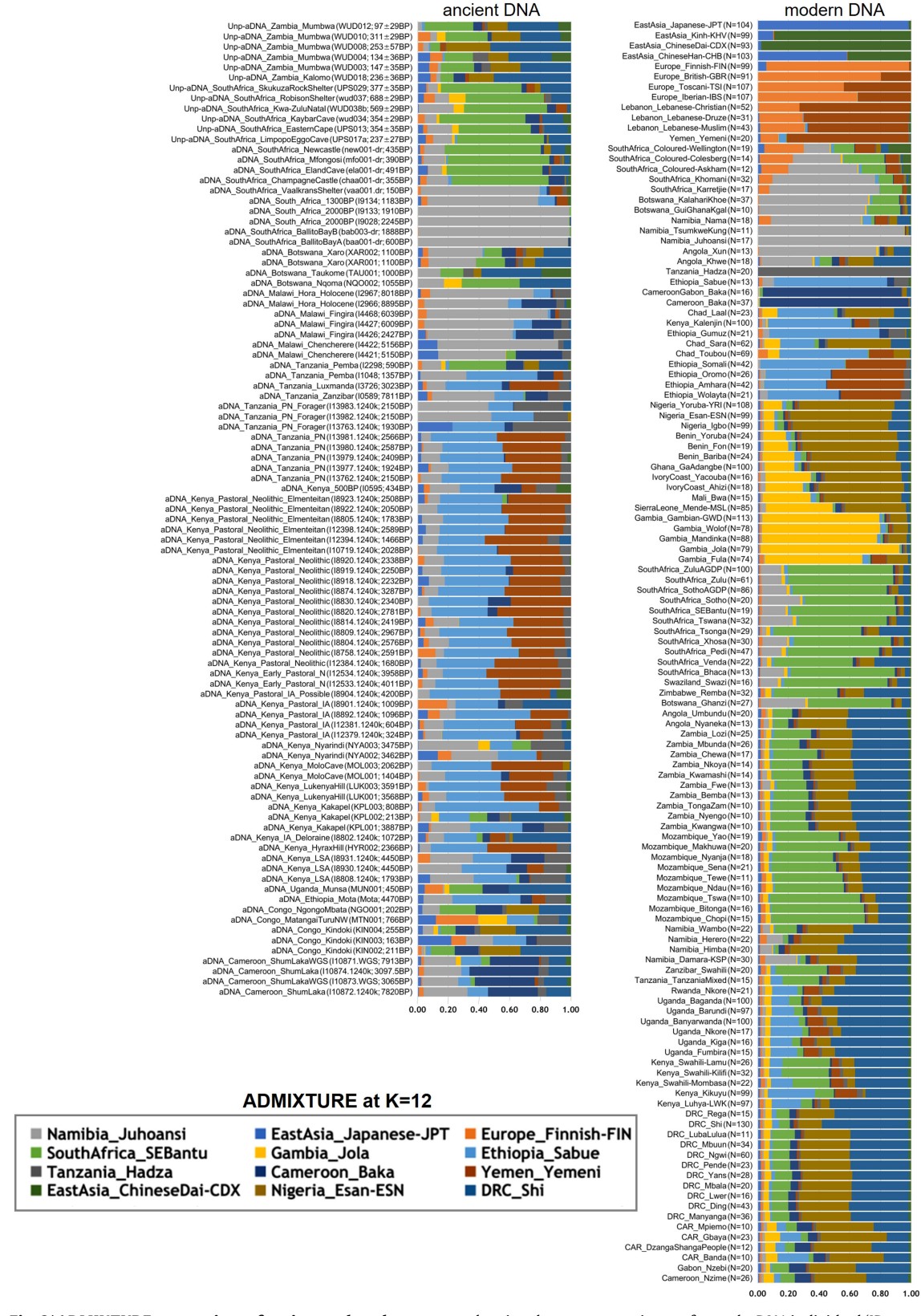

**Extended Data Fig. 8 | ADMIXTURE proportions of ancient and modern populations.** ADMIXTURE proportions of ancient and modern African and Eurasian populations estimated for K = 12 ancestral components. Figure showing the ancestry estimates for each aDNA individual (ID sample and date are indicated in parenthesis) and the ancestry averages across individuals for each modern population (sample sizes are indicated in parenthesis).

**Extended Data Table 1 | Genetic affinity of BSP to non-BSP estimated using f3-statistics**

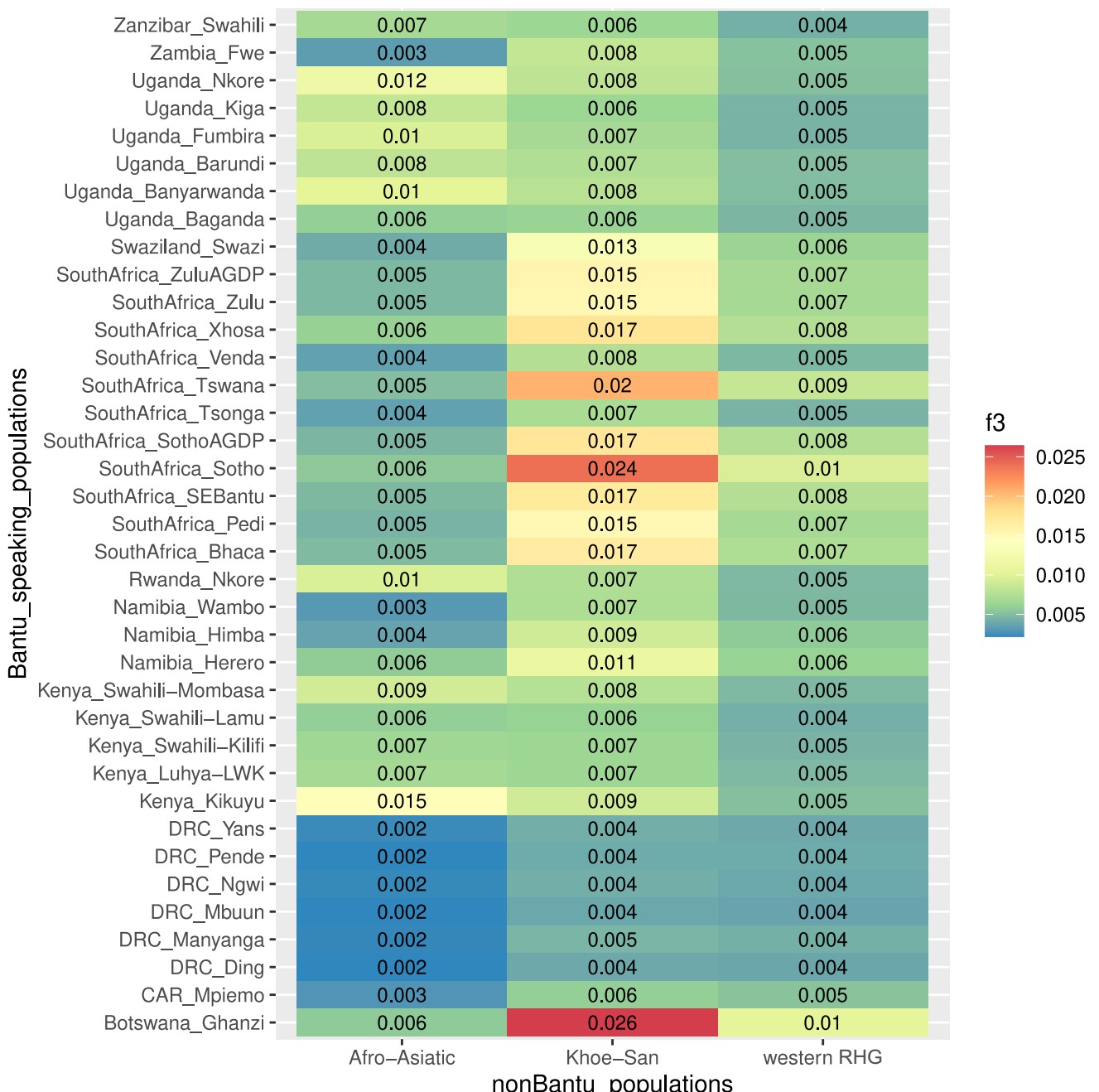

Genetic admixture was tested using *f*3-statistics in the form *f*3(Yoruba; non-BSP, target BSP). The comparative non-BSP are represented by an Afro-Asiatic-speaking population from Ethiopia (Amhara), a Khoe-San population (Ju/'hoansi) and the western RHG Baka population. The figure shows the mean values for each population. The standard errors (SE) of each mean are reported in Supplementary Figs. 28–30.

## Extended Data Table 2 | Genetic affinity of ancient individuals to modern BSP

| Bantu-speaking populations | UPS013 | UPS017a | UPS029 | WUD034 | WUD037 | WUD038b | WUD003 | WUD004 | WUD008 | WUD010 | WUD012 | WUD018 |
|---|---|---|---|---|---|---|---|---|---|---|---|---|
| Zimbabwe_Remba | 0.005 | 0.006 | 0.006 | 0.006 | 0.006 | 0.006 | 0.004 | 0.003 | 0.005 | 0.003 | 0.004 | 0.004 |
| Zanzibar_Swahili | 0.006 | 0.005 | 0.005 | 0.006 | 0.005 | 0.005 | 0.004 | 0.002 | 0.006 | 0.004 | 0.005 | 0.004 |
| Zambia_TongaZam | 0.005 | 0.004 | 0.004 | 0.003 | 0.005 | 0.005 | 0.004 | 0.009 | 0.005 | 0.004 | 0.006 | 0.003 |
| Zambia_Nyengo | 0.004 | 0.005 | 0.006 | 0.003 | 0.005 | 0.004 | 0.005 | 0.005 | 0.001 | 0.005 | 0.006 | 0.008 |
| Zambia_Nkoya | 0.004 | 0.004 | 0.003 | 0.004 | 0.003 | 0.004 | 0.005 | 0.006 | 0.005 | 0.003 | 0.003 | 0.005 |
| Zambia_Mbunda | 0.004 | 0.004 | 0.003 | 0.004 | 0.004 | 0.003 | 0.003 | 0.001 | 0.004 | 0.003 | 0.004 | 0.005 |
| Zambia_Lozi | 0.005 | 0.004 | 0.004 | 0.005 | 0.005 | 0.005 | 0.003 | 0.004 | 0.005 | 0.004 | 0.004 | 0.004 |
| Zambia_Kwangwa | 0.004 | 0.005 | 0.005 | 0.004 | 0.005 | 0.004 | 0.004 | 0.001 | 0.005 | 0.005 | 0.005 | 0.003 |
| Zambia_Kwamashi | 0.004 | 0.004 | 0.004 | 0.004 | 0.004 | 0.004 | 0.004 | 0.004 | 0.002 | 0.004 | 0.004 | 0.003 |
| Zambia_Fwe | 0.005 | 0.005 | 0.005 | 0.004 | 0.005 | 0.005 | 0.004 | 0.003 | 0.004 | 0.003 | 0.004 | 0.007 |
| Zambia_Chewa | 0.005 | 0.005 | 0.005 | 0.005 | 0.004 | 0.004 | 0.004 | 0.004 | 0.006 | 0.004 | 0.004 | 0.003 |
| Zambia_Bemba | 0.004 | 0.004 | 0.005 | 0.003 | 0.006 | 0.004 | 0.004 | 0.005 | 0.003 | 0.004 | 0.005 | 0.006 |
| Uganda_Nkore | 0.005 | 0.005 | 0.004 | 0.005 | 0.003 | 0.004 | 0.004 | 0.006 | 0.007 | 0.004 | 0.004 | 0.004 |
| Uganda_Kiga | 0.005 | 0.004 | 0.004 | 0.004 | 0.003 | 0.004 | 0.003 | 0.005 | 0.005 | 0.003 | 0.004 | 0.004 |
| Uganda_Fumbira | 0.005 | 0.005 | 0.004 | 0.004 | 0.004 | 0.004 | 0.004 | 0.006 | 0.005 | 0.004 | 0.004 | 0.003 |
| Uganda_Barundi | 0.004 | 0.004 | 0.004 | 0.004 | 0.004 | 0.004 | 0.004 | 0.005 | 0.006 | 0.004 | 0.004 | 0.004 |
| Uganda_Banyarwanda | 0.005 | 0.004 | 0.004 | 0.004 | 0.004 | 0.005 | 0.004 | 0.005 | 0.005 | 0.004 | 0.004 | 0.005 |
| Uganda_Baganda | 0.004 | 0.004 | 0.004 | 0.005 | 0.004 | 0.004 | 0.004 | 0.003 | 0.005 | 0.004 | 0.004 | 0.004 |
| Tanzania_TanzaniaMixed | 0.005 | 0.005 | 0.005 | 0.006 | 0.005 | 0.005 | 0.004 | 0.005 | 0.006 | 0.004 | 0.006 | 0.005 |
| Swaziland_Swazi | 0.008 | 0.009 | 0.008 | 0.009 | 0.01 | 0.009 | 0.004 | 0.006 | 0.005 | 0.005 | 0.006 | 0.003 |
| SouthAfrica_ZuluAGDP | 0.009 | 0.009 | 0.008 | 0.009 | 0.009 | 0.01 | 0.005 | 0.004 | 0.005 | 0.005 | 0.005 | 0.005 |
| SouthAfrica_Zulu | 0.008 | 0.009 | 0.008 | 0.009 | 0.009 | 0.01 | 0.005 | 0.004 | 0.004 | 0.005 | 0.005 | 0.005 |
| SouthAfrica_Xhosa | 0.008 | 0.009 | 0.008 | 0.01 | 0.009 | 0.011 | 0.005 | 0.004 | 0.005 | 0.006 | 0.005 | 0.005 |
| SouthAfrica_Venda | 0.006 | 0.007 | 0.007 | 0.007 | 0.006 | 0.006 | 0.005 | 0.004 | 0.004 | 0.004 | 0.004 | 0.004 |
| SouthAfrica_Tswana | 0.01 | 0.009 | 0.008 | 0.009 | 0.01 | 0.01 | 0.005 | 0.005 | 0.007 | 0.005 | 0.006 | 0.005 |
| SouthAfrica_Tsonga | 0.006 | 0.007 | 0.007 | 0.007 | 0.008 | 0.007 | 0.004 | 0.005 | 0.007 | 0.005 | 0.005 | 0.004 |
| SouthAfrica_SothoAGDP | 0.009 | 0.009 | 0.008 | 0.009 | 0.008 | 0.01 | 0.005 | 0.006 | 0.005 | 0.005 | 0.005 | 0.005 |
| SouthAfrica_Sotho | 0.01 | 0.011 | 0.009 | 0.011 | 0.011 | 0.01 | 0.005 | 0.005 | 0.006 | 0.005 | 0.006 | 0.004 |
| SouthAfrica_SEBantu | 0.009 | 0.01 | 0.008 | 0.009 | 0.01 | 0.011 | 0.005 | 0.004 | 0.006 | 0.006 | 0.006 | 0.005 |
| SouthAfrica_Pedi | 0.008 | 0.009 | 0.008 | 0.009 | 0.008 | 0.009 | 0.005 | 0.004 | 0.005 | 0.005 | 0.005 | 0.005 |
| SouthAfrica_Bhaca | 0.008 | 0.009 | 0.008 | 0.01 | 0.009 | 0.011 | 0.005 | 0.004 | 0.007 | 0.005 | 0.005 | 0.006 |
| Rwanda_Nkore | 0.005 | 0.004 | 0.004 | 0.004 | 0.004 | 0.004 | 0.003 | 0.003 | 0.006 | 0.004 | 0.003 | 0.004 |
| Namibia_Wambo | 0.004 | 0.004 | 0.004 | 0.004 | 0.004 | 0.005 | 0.003 | 0.006 | 0.003 | 0.004 | 0.004 | 0.003 |
| Namibia_Himba | 0.005 | 0.005 | 0.005 | 0.004 | 0.005 | 0.004 | 0.004 | 0.006 | 0.006 | 0.004 | 0.004 | 0.005 |
| Namibia_Herero | 0.005 | 0.004 | 0.004 | 0.005 | 0.005 | 0.005 | 0.004 | 0.009 | 0.006 | 0.005 | 0.003 | 0.004 |
| Mozambique_Yao | 0.007 | 0.007 | 0.007 | 0.008 | 0.008 | 0.006 | 0.005 | 0.006 | 0.005 | 0.005 | 0.005 | 0.003 |
| Mozambique_Tswa | 0.007 | 0.008 | 0.007 | 0.007 | 0.009 | 0.006 | 0.005 | 0.004 | 0.004 | 0.005 | 0.005 | 0.006 |
| Mozambique_Tewe | 0.006 | 0.006 | 0.006 | 0.007 | 0.007 | 0.007 | 0.004 | 0.005 | 0.005 | 0.006 | 0.005 | 0.005 |
| Mozambique_Sena | 0.006 | 0.007 | 0.007 | 0.006 | 0.006 | 0.006 | 0.005 | 0.004 | 0.007 | 0.005 | 0.006 | 0.005 |
| Mozambique_Nyanja | 0.005 | 0.006 | 0.005 | 0.005 | 0.005 | 0.006 | 0.004 | 0.002 | 0.003 | 0.004 | 0.005 | 0.005 |
| Mozambique_Ndau | 0.006 | 0.007 | 0.007 | 0.006 | 0.006 | 0.006 | 0.005 | 0.006 | 0.006 | 0.006 | 0.004 | 0.004 |
| Mozambique_Makhuwa | 0.006 | 0.007 | 0.006 | 0.007 | 0.005 | 0.007 | 0.005 | 0.005 | 0.007 | 0.005 | 0.006 | 0.004 |
| Mozambique_Chopi | 0.008 | 0.009 | 0.008 | 0.009 | 0.009 | 0.009 | 0.006 | 0.007 | 0.006 | 0.006 | 0.006 | 0.005 |
| Mozambique_Bitonga | 0.007 | 0.008 | 0.007 | 0.007 | 0.007 | 0.007 | 0.005 | 0.006 | 0.005 | 0.006 | 0.004 | 0.004 |
| Kenya_Swahili–Mombasa | 0.007 | 0.007 | 0.006 | 0.006 | 0.006 | 0.007 | 0.006 | 0.006 | 0.008 | 0.006 | 0.007 | 0.005 |
| Kenya_Swahili–Lamu | 0.005 | 0.005 | 0.005 | 0.006 | 0.005 | 0.005 | 0.004 | 0.004 | 0.006 | 0.004 | 0.006 | 0.005 |
| Kenya_Swahili–Kilifi | 0.005 | 0.006 | 0.005 | 0.006 | 0.005 | 0.005 | 0.004 | 0.005 | 0.006 | 0.004 | 0.004 | 0.005 |
| Kenya_Luhya–LWK | 0.005 | 0.005 | 0.004 | 0.004 | 0.004 | 0.004 | 0.004 | 0.004 | 0.006 | 0.003 | 0.004 | 0.005 |
| Kenya_Kikuyu | 0.005 | 0.005 | 0.005 | 0.005 | 0.006 | 0.005 | 0.004 | 0.006 | 0.006 | 0.004 | 0.005 | 0.005 |
| Kenya_Kalenjin | 0.005 | 0.004 | 0.004 | 0.004 | 0.004 | 0.005 | 0.004 | 0.005 | 0.005 | 0.003 | 0.004 | 0.004 |
| Gabon_Nzebi | 0.004 | 0.004 | 0.003 | 0.003 | 0.005 | 0.003 | 0.004 | 0.003 | 0.004 | 0.003 | 0.003 | 0.004 |
| DRC_Yans | 0.003 | 0.003 | 0.003 | 0.003 | 0.002 | 0.003 | 0.003 | 0.005 | 0.005 | 0.003 | 0.002 | 0.002 |
| DRC_Shi | 0.004 | 0.004 | 0.004 | 0.004 | 0.004 | 0.003 | 0.003 | 0.003 | 0.005 | 0.003 | 0.003 | 0.004 |
| DRC_Rega | 0.004 | 0.004 | 0.003 | 0.004 | 0.003 | 0.004 | 0.004 | −0.001 | 0.006 | 0.003 | 0.004 | 0.003 |
| DRC_Pende | 0.004 | 0.003 | 0.003 | 0.004 | 0.002 | 0.004 | 0.003 | 0.002 | 0.004 | 0.003 | 0.004 | 0.002 |
| DRC_Ngwi | 0.003 | 0.003 | 0.003 | 0.003 | 0.003 | 0.003 | 0.003 | 0.004 | 0.005 | 0.003 | 0.004 | 0.003 |
| DRC_Mbuun | 0.003 | 0.002 | 0.003 | 0.003 | 0.003 | 0.003 | 0.003 | 0.002 | 0.004 | 0.003 | 0.003 | 0.003 |
| DRC_Mbala | 0.003 | 0.003 | 0.003 | 0.003 | 0.003 | 0.003 | 0.003 | 0.005 | 0.003 | 0.003 | 0.003 | 0.004 |
| DRC_Manyanga | 0.004 | 0.004 | 0.004 | 0.004 | 0.004 | 0.003 | 0.003 | 0.003 | 0.003 | 0.003 | 0.004 | 0.003 |
| DRC_Lwer | 0.003 | 0.003 | 0.003 | 0.004 | 0.003 | 0.002 | 0.003 | 0.005 | 0.004 | 0.002 | 0.003 | 0.004 |
| DRC_LubaLulua | 0.003 | 0.003 | 0.003 | 0.004 | 0.004 | 0.003 | 0.003 | 0.003 | 0.005 | 0.003 | 0.003 | 0.005 |
| DRC_Ding | 0.003 | 0.003 | 0.004 | 0.003 | 0.002 | 0.003 | 0.003 | 0.003 | 0.005 | 0.003 | 0.003 | 0.003 |
| CAR_Mpiemo | 0.003 | 0.003 | 0.003 | 0.004 | 0.001 | 0.003 | 0.003 | 0.004 | 0.004 | 0.003 | 0.003 | 0.006 |
| CAR_Gbaya | 0.002 | 0.002 | 0.002 | 0.002 | 0.003 | 0.001 | 0.002 | 0 | 0.002 | 0.001 | 0.002 | 0.002 |
| CAR_DzangaShangaPeople | 0.003 | 0.003 | 0.003 | 0.003 | 0.003 | 0.003 | 0.003 | 0.001 | 0.002 | 0.003 | 0.004 | 0.004 |
| CAR_Banda | 0.002 | 0.002 | 0.002 | 0.003 | 0.002 | 0.003 | 0.002 | 0.001 | 0.006 | 0.003 | 0.004 | 0.002 |
| Cameroon_Nzime | 0.003 | 0.003 | 0.003 | 0.004 | 0.003 | 0.003 | 0.003 | 0.003 | 0.004 | 0.004 | 0.003 | 0.003 |
| Botswana_Ghanzi | 0.011 | 0.01 | 0.009 | 0.01 | 0.011 | 0.01 | 0.006 | 0.006 | 0.006 | 0.006 | 0.006 | 0.004 |
| Angola_Umbundu | 0.005 | 0.005 | 0.005 | 0.005 | 0.005 | 0.005 | 0.004 | 0.005 | 0.004 | 0.006 | 0.006 | 0.006 |
| Angola_Nyaneka | 0.004 | 0.004 | 0.005 | 0.004 | 0.003 | 0.003 | 0.004 | 0.003 | 0.003 | 0.003 | 0.003 | 0.003 |

f3

0.009
0.006
0.003
0.000

**South African individuals** (UPS013, UPS017a, UPS029, WUD034, WUD037, WUD038b)
**Zambian individuals** (WUD003, WUD004, WUD008, WUD010, WUD012, WUD018)

Genetic affinities were estimated using the *f*3-statistic, in the form f3(Yoruba; ancient sample, BSP). Positive values of *f*3 indicate increasing genetic affinity between modern BSP and each ancient individual. The first six individuals are from current-day South Africa (UPS013, UPS017a, UPS029, WUD034, WUD037 and WUD038b) and the remaining six individuals from current-day Zambia (WUD003, WUD004, WUD008, WUD010, WUD012 and WUD018). Radiocarbon calibrated ages and genome statistics are reported in Supplementary Table 14. All South African ancient individuals show greater affinity to current-day BSP, whereas Zambian ancient individuals show less clear affinity patterns, supporting the notion that current-day Zambia was a crossroad of interaction.

# Reporting Summary

## Statistics

For all statistical analyses, confirm that the following items are present in the figure legend, table legend, main text, or Methods section.

| n/a | Confirmed | |
|---|---|---|
| ☐ | ☒ | The exact sample size (*n*) for each experimental group/condition, given as a discrete number and unit of measurement |
| ☐ | ☒ | A statement on whether measurements were taken from distinct samples or whether the same sample was measured repeatedly |
| ☐ | ☒ | The statistical test(s) used AND whether they are one- or two-sided *Only common tests should be described solely by name; describe more complex techniques in the Methods section.* |
| ☒ | ☐ | A description of all covariates tested |
| ☒ | ☐ | A description of any assumptions or corrections, such as tests of normality and adjustment for multiple comparisons |
| ☐ | ☒ | A full description of the statistical parameters including central tendency (e.g. means) or other basic estimates (e.g. regression coefficient) AND variation (e.g. standard deviation) or associated estimates of uncertainty (e.g. confidence intervals) |
| ☐ | ☒ | For null hypothesis testing, the test statistic (e.g. *F*, *t*, *r*) with confidence intervals, effect sizes, degrees of freedom and *P* value noted *Give P values as exact values whenever suitable.* |
| ☒ | ☐ | For Bayesian analysis, information on the choice of priors and Markov chain Monte Carlo settings |
| ☒ | ☐ | For hierarchical and complex designs, identification of the appropriate level for tests and full reporting of outcomes |
| ☒ | ☐ | Estimates of effect sizes (e.g. Cohen's *d*, Pearson's *r*), indicating how they were calculated |

*Our web collection on statistics for biologists contains articles on many of the points above.*

## Software and code

Policy information about availability of computer code

| | |
|---|---|
| Data collection | OxCal v4.4, ContamMix, SHCal20, HaploGrep2, GenomeStudio v2.0.3, PLINK v1.9 |
| Data analysis | ADMIXTOOLS v2, Eigensoft v7.2.1, ADMIXTURE v1.3.0, SHAPEIT v2.r904, RFMix v1.5.4, MOSAIC v1.4, KIN v1.4, IBDNe (ibdne.19Sep19.268.jar), PLINK v1.9, SpaceMix v0.13, ASCEND v10, Beagle v4.1, TreeMix v1.13, in-house R and Python scripts (were included here: https://github.com/Schlebusch-lab/Expansion_of_BSP_peer-reviewed_article), EEMS, FEEMS, GenGrad. |

For manuscripts utilizing custom algorithms or software that are central to the research but not yet described in published literature, software must be made available to editors and reviewers. We strongly encourage code deposition in a community repository (e.g. GitHub). See the Nature Portfolio guidelines for submitting code & software for further information.

## Data

Policy information about availability of data

All manuscripts must include a data availability statement. This statement should provide the following information, where applicable:
- Accession codes, unique identifiers, or web links for publicly available datasets
- A description of any restrictions on data availability
- For clinical datasets or third party data, please ensure that the statement adheres to our policy

Novel genome-wide genotype data of modern-day African populations (*.tped and *.tfam files) and whole-genome data of aDNA individuals (*.bam files) generated in this study will be made available through the European Genome-phenome Archive (EGA) data repository (EGA accessory numbers: EGAS00001007519 and

EGAS00001007515). Controlled access policies guided by participant consent agreements will be implemented by the AfricanNeo Data Access Committee (DAC accessory number: EGAC00001003398).

# Research involving human participants, their data, or biological material

Policy information about studies with human participants or human data. See also policy information about sex, gender (identity/presentation), and sexual orientation and race, ethnicity and racism.

| | |
|---|---|
| Reporting on sex and gender | Biological samples were obtained from any participants of both sex, and sex or gender were not a factor in the sampling collection. |
| Reporting on race, ethnicity, or other socially relevant groupings | New samples presented in this study were collected in large-scale sampling campaigns conducted in fourteen sub-Saharan African countries (1,763 individuals in total). Participants were recruited based on self identification as member of a specific ethno-linguist group in Africa. |
| Population characteristics | Biological samples were collected from healthy adults. |
| Recruitment | All the individuals who participated in sample collection provided informed consent. |
| Ethics oversight | Ethical permits and sampling permission were obtained in African countries and the study as a whole was approved by the Swedish ethical review board (DNR-2021-01448). Granted ethics was approval by: the Human Research Ethics Committee (Medical) (University of the Witwatersrand, South Africa; protocol Nr. M180656); the Biomedical Research Ethics Board (University of Zambia,Zambia; protocol number: 004-08-07); the Faculty of Natural and Agricultural Sciences Ethics Committee (University of Pretoria, South Africa; protocol number: EC160429-024 and 259/2016); the Swedish National Ethics Committee (Sweden; protocol number: Dnr 2019-05244), and the Minister of Arts and Culture (DRC; protocol number: Nr 091/CAB/MIN/CA/PKB/2018). |

Note that full information on the approval of the study protocol must also be provided in the manuscript.

# Field-specific reporting

Please select the one below that is the best fit for your research. If you are not sure, read the appropriate sections before making your selection.

☒ Life sciences ☐ Behavioural & social sciences ☐ Ecological, evolutionary & environmental sciences

For a reference copy of the document with all sections, see nature.com/documents/nr-reporting-summary-flat.pdf

# Life sciences study design

All studies must disclose on these points even when the disclosure is negative.

| | |
|---|---|
| Sample size | Our analyses are based on the data we collected from modern and ancient individuals. Standard errors are reported to describe ranges of our analyses. |
| Data exclusions | We exclude individuals with low SNP-genotyping rates for modern samples or low coverage for aDNA individuals. |
| Replication | We replicated estimates for admixture patterns by running ADMIXTURE analyses for each K-group by using 10 independent runs, with a random seed for each K-group. Results were not externally replicated. |
| Randomization | We used randomization for our simulations. |
| Blinding | Blinding was not applicable in this study. The study design did not allocate samples to specific groups such as "cases" or "controls". |

# Reporting for specific materials, systems and methods

We require information from authors about some types of materials, experimental systems and methods used in many studies. Here, indicate whether each material, system or method listed is relevant to your study. If you are not sure if a list item applies to your research, read the appropriate section before selecting a response.

## Materials & experimental systems

| n/a | Involved in the study |
|-----|------------------------|
| ☒ | ☐ Antibodies |
| ☒ | ☐ Eukaryotic cell lines |
| ☐ | ☒ Palaeontology and archaeology |
| ☒ | ☐ Animals and other organisms |
| ☒ | ☐ Clinical data |
| ☒ | ☐ Dual use research of concern |
| ☒ | ☐ Plants |

## Methods

| n/a | Involved in the study |
|-----|------------------------|
| ☒ | ☐ ChIP-seq |
| ☒ | ☐ Flow cytometry |
| ☒ | ☐ MRI-based neuroimaging |

# Palaeontology and Archaeology

Specimen provenance

The 12 new ancient human remains in this study came from various caves and rock shelters in Zambia and South Africa. We obtained permission from the South African Heritage Resources Agency (SAHRA) to sample and export bones for ancient DNA analyses. Nine WUD samples (permit number: 2789) are from the Raymond A. Dart Archaeological Human Remains Collection (Dart Collection) located at the School of Anatomical Sciences, University of the Witwatersrand (Johannesburg, South Africa). Three UPS samples (permit number: 2804) are from the Archaeological Human remains Collection (Pretoria Bone Collection) situated within the Department of Anatomy, University of Pretoria (Pretoria, South Africa).

For both collections Prof. M. Steyn is the permit holder. The archeological context, morphological assessments, and dating of the remains were described before for six of the samples: WUD034 and WUD037 (C1 and C9 in Meyer et al. 2021; and WUD003, WUD004, WUD008, and WUD010 (Steyn et al 2023). WUD038b sample originated from an archeological site in KwaZulu Natal but is curated in the Dart Collection. WUD012 (Chipongwe Caves) was collected in 1930 by Raymond Dart. WUD012 and WUD018 were originally collected in current-day Zambia and are curated in the Dart Collection, while UPS013, UPS017a, and UPS029 are kept in the Pretoria Bone Collection. Little is known about their archeological contexts.

Specimen deposition

Archeological human remains are housed in the Raymond A. Dart Archaeological Human Remains Collection and the University of Pretoria Bone Collection.

Dating methods

Six samples (WUD038b, WUD012, WUD018, UPS013, UPS017a, and UPS029) were accelerator mass spectrometry (AMS) radiocarbon dated at the Tandem Laboratory (Department of Physics and Astronomy, Uppsala University, Sweden). Radiocarbon dates were calibrated with OxCal 4.4 38 using the atmospheric curve SHCal20 39 and are given at 95.4% probability (2σ).

☐ Tick this box to confirm that the raw and calibrated dates are available in the paper or in Supplementary Information.

Ethics oversight

We obtained all the permissions necessary for ancient DNA analyses from the respective countries.

Note that full information on the approval of the study protocol must also be provided in the manuscript.

