## [Peer Review File · Nature]

Manuscript Title: The genetic legacy of the expansion of Bantu-speaking peoples in Africa

Reviewer Comments & Author Rebuttals

Reviewer Reports on the Initial Version:

Referees' comments:

Referee #1 (Remarks to the Author):

The authors collected and genotyped an impressive number of novel African genomes (both modern and ancient) which complement the current picture available from African populations in an anthropological/cultural/geographic/linguistic informed way. The resource they provide, along with the additional genomes gathered from the literature, will be of great use for future studies and as a reference for evolutionary and biomedical studies in Africa.

With their thorough analyses and carefully thought study design, the authors also managed to provide novel and convincing answers to the long standing debate about the routes and impact on the African genetic landscape of the so called Bantu expansion.

I think the article is basically almost ready for publication in its current form (clarity, robustness, reliability etc are all vastly met), and I would have just two minor additional analyses to suggest (provided they are not already present in the deluge of supplementary information and that I accidentally overlooked them). Both points concern the downstream analyses following the local ancestry deconvolution of the Bantu component away from the other African and Eurasian genetic components:

1) The authors describe the phasing and imputation procedure that followed the extraction of the Bantu components in order to minimize missing data. This procedure is surely well justified, however it may introduce biases which can easily be controlled for: for example, the authors could compare the PCA obtained after projecting the Bantu-masked genomes with no further phasing/imputation onto an African genomic landscape (say, using genomes for which at least 50% of the genome is in Bantu status) and see how this PCA compares with the phased/imputed one.

2) Given the availability of masked haplotypes also from non-Bantu components (South African HG, Central African HG, Afroasiatic and Eurasian), it would be nice to see at least a PCA computed by projecting these masked genomes (without phasing/imputation, as per my previous point and to minimize the computational burden during the review process) onto a PCA computed using just South African HG, or just Central African HG or just Afroasiatic or just Eurasian genomes, respectively. By this way it would be possible to at least in part retrieve the pre-Bantu genetic landscape of the populations affected by the expansion and/or to trace further connections among them in the aftermaths of the expansion.

Referee #2 (Remarks to the Author):

Fortes-Lima et al present an impressive genetic analysis of Bantu-Speaking populations sampled throughout Sub-Saharan Africa. Their analysis includes 1,740 newly genotyped modern samples as well as 12 newly sequenced ancient DNA samples. The size and diversity of this dataset allows them to thoroughly investigate the demographic history and genetic diversity of Bantu-Speaking populations. They use a large collection of population genetic/bioinformatics methods to address questions regarding population structure, admixture with non-Bantu-Speaking populations, and

the migration route(s) that the Bantu expansion likely followed. Their analyses reveal that many Bantu-Speaking populations have differential patterns of admixture with non-Bantu-Speaking populations, with the different admixture sources mainly coming from geographically proximate groups. Measures of genetic diversity indicate a serial founder effect where genetic diversity decreases with increasing distance from the predicted origin of the Bantu Expansion. Through spatial based modeling they provide additional support for the Bantu expansion following the late-split hypothesis. Additional modeling presents evidence of migration barriers and interaction zones between Bantu-Speaking populations, although the authors note that further research is required to refine additional aspects of these models. The paper is well written and many of the conclusions are well supported by their analyses, making this paper an important contribution to the study of human evolution. This dataset will also be an important resource for future studies of sub-Saharan African genetic diversity. I do have comments regarding the technical details on some methods that would help make some conclusions/analyses in their paper more clear/robust.

Major Comments:

1. The IBDNe analyses that show recent changes in effective population size are interesting. However, there are a few factors that are not discussed in the manuscript, but should be considered when interpreting these results. How does the lower sample size of each population impact these results? It is generally best to run IBDNe with 100s of samples/population. Are the confidence intervals wide, especially in the most recent time frames for populations with small sample sizes? IBDNe can also be impacted by admixture, which is inferred for many of the populations in this dataset. How robust are your IBDNe results to the types of gene flow inferred for these populations?
2. Local ancestry inference:
 - a. What parameter of generations since admixture was used in RFMix analysis and how was this parameter chosen?
 - b. Do the estimates of non-BSP RFMix modeled local ancestry correlate well with the non-BSP ADMIXTURE modeled ancestry estimates? If they are drastically different, could this indicate that you under/over masked certain individuals/populations?
 - c. I could not find Figure S4.4 and S4.5 (cited on page 4 in the main text) in the supplement. The supplement goes from Fig.S 4.3 to Fig.S 5.1.
3. EEMS analysis:
 - a. Would an approach that uses identity-by-descent segments for use in EEMS like in Al-Asadi et al (2019, PMID: 30640906), be helpful for further analysis of the possible barriers to migration in the proposed interaction zones presented in Figure 5? This might help assess if the migration barriers estimated by EEMS were consistent throughout the past ~75-100 generations or changed over time.
 - b. Could the migration barrier inferred between the populations in South Africa/Mozambique and the population in Botswana (Figure 5c & S10.7c) be impacted by the sparse geographic sampling between these locations? I find this migration barrier interesting because Figure S10.7d shows that these populations have similar ADMIXTURE profiles. Could this also apply to why the barrier to migration is found in Zambia/DRC compared to the high migration zone found on the Indian Ocean coast where the geographic sampling density is higher?
4. On page 4 it is noted that the Herero and Himba show especially strong signals of founder effects compared to the rest of the Bantu speaking populations. It is reasoned that this could be due to "the consequence of genetic isolation since their arrival in southwestern Africa and recent endogamic practices linked to cattle herding." Are there additional explanations or factors that could be impacting this? I am specifically thinking of the Herero genocide in the early 1900's. Could this also be contributing to the signals of high ROH and high inbreeding coefficient? Or is the early 1900's too early of a time frame for this to impact your results?
5. Figure S3.4 shows that K=16 has the lowest cross validation, yet Figure 2a presents the ADMIXTURE results for K=12. Why was K=12 chosen to display in the main text instead of K=16?

Minor Comments

6. It looks like the setup of the F3 statistic is in the form of the out-group F3, where larger values

indicate the two populations are more closely related. If you flipped the F3 in Figure S3.16 to be f3(Target; Yoruba, Amhara) would the values that are much greater than 0 in the current figure flip to being negative, therefore indicating that the Target has admixture between the Yoruba and Amhara?

7. Has the data upload process/application already been started with the European Genome-phenome Archive for the novel data?

Referee #3 (Remarks to the Author):

This paper reports a large genetic dataset of Bantu speaking populations (BSP) throughout Africa. It provides many descriptive analyses of how genetic similarities and diversity among BSP decline with time and geographic distance from their presumed west African origins. It provides some analyses of probable routes of migration as the Bantu swept out of West Africa around 5kya eventually ending up in present day South Africa.

These questions have been investigated by previous authors and this poses some difficulties for the current paper in identifying what is new about their contribution. Two papers using linguistic and geographical data (Grollemund et al 2015 and Koile et al 2022), both of which the authors cite, provide detailed analyses of the timings and routes of the Bantu migration, and the current paper does not add any new insights to these (a third previous paper – Currie et al, 2013 – does not provide dates but mostly agrees on the route of migration).

In fact, the current paper suggests a date in the abstract for the origin of the Bantu expansion (the Bantu ingroup) that is in my view too young, and it is at odds with the two previous papers above, both of which find close agreement on this date. The authors do not provide any reason to prefer their suggested date over that of previous studies, and I might have missed it but I could not even find any direct evidence about dates in the paper or the 120 pages of SI. There were some materials on dates in Table S.7 but they are given in generations and I was not able to relate them to the trees they presented in the Supplementary Figures.

The same points could be made about the proposed migration routes. The previous papers broadly agree on routes while differing somewhat on one of the early timings. The current paper does not provide any grounds for preferring one of the previous works over the other, neither does it provide credible new routes.

The current paper asks whether there was genetic interbreeding between BSP and non-BSP groups along the way and asserts that "We show for the first time that genetic diversity amongst Bantu-speaking populations declines with distance from western Africa". This is misleading. Both previous genetic studies of the Bantu that the authors cite (authors' references 22 and 23) conclude that there has been substantial admixture between BSP and non-BSP people at a variety of sites around Africa. Indeed, this admixture is very well known among geneticists because it has confounded for a long time attempts to study contemporary hunter-gatherer populations in southern Africa such as the San people, or among groups like the Sandawe or Hadza in East Africa.

Now, admixture alone does not entail loss of diversity, but admixture does point towards relatively small migrant groups moving into occupied territories, rather than a huge invading force that swamped or drove the local inhabitants out. And the authors find a number of demographic founder events. Small groups, of course, lose diversity via sampling and so showing here that the Bantu genetic diversity declines with distance is a very minor addition, especially as the slope of the line is shallow, and this finding is not used to make points about social structure or perhaps adaptation.

The authors study the admixture further and conclude that the pattern of results that describes how Bantu populations' similarity to their forebearers declines with time and space supports a serial founder effects model over a simpler isolation by distance model. I am prepared to believe that it might, but the authors do not distinguish this result from a simpler bottleneck model.

Even if they could distinguish the two, this (serial founder effect) is a minor statement and the authors do not link it to anything important anthropologically or archaeologically. If people did pause, why, what were the implications, and what made them important. For example, the Grollemund et al paper finds empirical evidence that when Bantu peoples moved into rainforest, they did so more slowly than when they moved within savannah. The interest here is that Bantu were farmers, accustomed to savannah but not to rainforest. Moving into the rainforest then probably required the acquisition of new knowledge and skills.

The presentation of results I'm afraid reads like a group of authors trying out the assembled toolkit of genetic analysis techniques and just reporting what they find. All of the figures are difficult to study (try, for instance, to distinguish among the colours in the legend of Figure 2), their implications are not well drawn, and the figure captions are not helpful. They have a bad habit of referring readers to supplementary figures and tables to aid understanding of their main text figures. But Figure captions need to stand alone. Figures 5a,b supposedly have four colours – I can only discern three – and these figures are not nearly as informative as comparable figures in Koile et al. or Grollemund et al.

Author Rebuttals to Initial Comments:

Referees' comments:

Referee #1 (Remarks to the Author):

The authors collected and genotyped an impressive number of novel African genomes (both modern and ancient) which complement the current picture available from African populations in an anthropological/cultural/geographic/linguistic informed way. The resource they provide, along with the additional genomes gathered from the literature, will be of great use for future studies and as a reference for evolutionary and biomedical studies in Africa.

With their thorough analyses and carefully thought study design, the authors also managed to provide novel and convincing answers to the long standing debate about the routes and impact on the African genetic landscape of the so called Bantu expansion.

I think the article is basically almost ready for publication in its current form (clarity, robustness, reliability etc are all vastly met), and I would have just two minor additional analyses to suggest (provided they are not already present in the deluge of supplementary information and that I accidentally overlooked them). Both points concern the downstream analyses following the local ancestry deconvolution of the Bantu component away from the other African and Eurasian genetic components:

We thank the reviewer for her/his suggestions and positive comments.

1) The authors describe the phasing and imputation procedure that followed the extraction of the Bantu components in order to minimize missing data. This procedure is surely well justified, however it may introduce biases which can easily be controlled for: for example, the authors could compare the PCA obtained after projecting the Bantu-masked genomes with no further phasing/imputation onto an African genomic landscape (say, using genomes for which at least 50% of the genome is in Bantu status) and see how this PCA compares with the phased/imputed one.

In the manuscript, we included the PCA before and after masking (with phasing and imputation) (Fig.S 13.2a and 13.2b). As the reviewer requested - we also include here (and in new Fig.S 13.2c) the masked BSP dataset (without phasing and imputation) projected onto the unmasked dataset of reference groups. As can be seen, there is very little difference between the phased/imputed dataset and the dataset without phasing/imputation when projecting the samples onto the genomic background of the reference groups. Imputation of missing data is, however, needed because in several analyses (e.g. non-projected PCA of only BSP groups, diversity statistics, EEMS analysis), only BSP groups are included and large blocks of non-overlapping missing data will influence analyses.

a) PCA plot of unmasked BSP and unmasked reference datasets (Fig.S 13.1). **b)** PCA plot of masked BSP (with 70% West Central African ancestry and **without** phasing and imputation) and unmasked reference datasets (new Fig.S 13.2). **c)** PCA plot of masked BSP (with 70% WCA and **after** phasing and imputation) and unmasked reference datasets.

2) Given the availability of masked haplotypes also from non-Bantu components (South African HG, Central African HG, Afroasiatic and Eurasian), it would be nice to see at least a PCA computed by projecting these masked genomes (without phasing/imputation, as per my previous point and to minimize the computational burden during the review process) onto a PCA computed using just South African HG, or just Central African HG or just Afroasiatic or just Eurasian genomes, respectively. By this way it would be possible to at least in part retrieve the pre-Bantu genetic landscape of the populations affected by the expansion and/or to trace further connections among them in the aftermaths of the expansion.

We also include Fig.S 13.3 in the manuscript where the non-Bantu haplotypes of BSP are projected for each ancestry onto the reference groups. Although some trends are visible that are different for each ancestry - we caution that we observed previously that ancestry assignments do not work well when shared ancestry with a parental group is below 70% ancestry. Below we show one example, more ancestries and PCs are available in Fig.S 13.3 in the Supplementary Material. As can be seen here, the East-African like BSP fragments are grouping in-between East African and West African groups on both PC1 and PC2 - this is probably due to over-masking. Similar (probable) over-masking trends are visible for other ancestries. However, as explained in response 2b to reviewer

2, our aim was to remove as much as possible non-Bantu admixture from genomes of BSP, and thus over-masking is a conservative scenario for our purpose. For this reason, we do not think that the removed fragments could be used for the purpose of finding the pre-Bantu landscape, as the reviewer suggests, since the fragments would be mixtures of overmasked West African ancestry and local ancestries. Thus if projected on South African HG, Central African HG or Afro-Asiatic groups, admixed West African ancestries within these groups would influence the location of the BSP removed haplotypes.

Fig.S 13.3a | AS-PCA plot showing East-African-like haplotypes of BSP (red dots) that are projected onto reference groups.

Referee #2 (Remarks to the Author):

Fortes-Lima et al present an impressive genetic analysis of Bantu-Speaking populations sampled throughout Sub-Saharan Africa. Their analysis includes 1,740 newly genotyped modern samples as well as 12 newly sequenced ancient DNA samples. The size and diversity of this dataset allows them to thoroughly investigate the demographic history and genetic diversity of Bantu-Speaking populations. They use a large collection of population genetic/bioinformatics methods to address questions regarding population structure, admixture with non-Bantu-Speaking populations, and the migration route(s) that the Bantu expansion likely followed. Their analyses reveal that many Bantu-Speaking populations have differential patterns of admixture with non-Bantu-Speaking populations, with the different admixture sources mainly coming from geographically proximate groups. Measures of genetic diversity indicate a serial founder effect where genetic diversity decreases with increasing distance from the predicted origin of the Bantu Expansion. Through spatial based modeling they provide additional support for the Bantu expansion following the late-split hypothesis. Additional modeling presents evidence of migration barriers and interaction zones between Bantu-Speaking populations, although the authors note that further research is required to refine additional aspects of these models. The paper is well written and many of the conclusions are well supported by their analyses, making this paper an important contribution to the study of human evolution. This dataset will also be an important resource for future studies of sub-Saharan African genetic diversity. I do have comments regarding the technical details on some methods that would help make some conclusions/analyses in their paper more clear/robust.

We thank the reviewer for her/his encouraging comments and suggestions.

Major Comments:

1. The IBDNe analyses that show recent changes in effective population size are interesting. However, there are a few factors that are not discussed in the manuscript, but should be considered when interpreting these results. How does the lower sample size of each population impact these results? It is generally best to run IBDNe with 100s of samples/population. Are the confidence intervals wide, especially in the most recent time frames for populations with small sample sizes? IBDNe can also be impacted by admixture, which is inferred for many of the populations in this dataset. How robust are your IBDNe results to the types of gene flow inferred for these populations?

We thank the reviewer for the feedback on our IBDNe analyses. We now include confidence intervals in all IBDNe figures for BSP from selected African regions (see new **Fig.S 6.1–6.3**).

We also agree with the reviewer that it is important to consider a sample size effect and admixture for IBDNe. We checked the effect of sample size here by estimating N_e for a few BSP with more than 40 individuals where their sample sizes were randomly downsampled to 10 (Pedi, Zulu, Ding, and SothoAGDP). As can be seen in the figure below, the sample size affects mainly recent generations and the range of the 95% confidence intervals (shaded area). However, the confidence intervals are not a good estimator of the accuracy of the estimated N_e in this approach. Since each CI is calculated using a bootstrap procedure with sample replacement (Browning and Browning, *AJHG* 2015), confidence intervals associated with lower sample sizes are expected to be lower than those associated with higher sample sizes.

From the examples below, we see that there is no systematic effect of downsampling. In some cases, the N_e trajectory is very similar to the one estimated with the full sample (Pedi and, for recent generations, Ding), while in others the N_e trajectory differs depending on the sample size (Zulu and SothoAGDP), indicating that it is not straightforward to predict the behavior of the method for lower

sample sizes. Therefore, we did not resample all our populations to lower (small) sample sizes. Instead, we included a figure that only has BSP with high sample sizes (>40) as new **Fig.S 6.1**. This serves as high confidence estimates of IBDNe. For these populations, we consider that the N_e is reasonably well estimated. We also added a caution to other IBDNe figures about the interpretation of the results for those populations with sample sizes lower than 40.

The effect of sample size on IBDNe results.

To investigate the effect of admixture on IBDNe segments we ran ancestry-specific (AS-)IBDNe on selected BSP from western, eastern, and southern Africa and included these as new **Fig.S 6.3**. We see that N_e estimates are generally higher for the whole population than for each ancestry taken separately, but the magnitude of this effect is different among populations, being very small in some cases. We added a caution remark to all IBDNe figure legends that admixture might have an effect on the estimations.

Fig.S 6.3 | Ancestry-specific (AS) IBDNe results for selected BSP.

2. Local ancestry inference:

a. What parameter of generations since admixture was used in RFMix analysis and how was this parameter chosen?

We used 25 generations ago (python RunRFMix.py “-G 25” option). This is the average number for the estimated admixture dates (Fig.S 11.1b). This is specified on page 11 of the Supplementary text (see Material and Methods point 1.9.): “We used 25 generations ago since the admixture event (“-G 25” option) because that was the median number obtained in the admixture timing analyses (see below)”.

b. Do the estimates of non-BSP RFMix modeled local ancestry correlate well with the non-BSP ADMIXTURE modeled ancestry estimates? If they are drastically different, could this indicate that you under/over masked certain individuals/populations?

As shown in previous studies (Martin et al. *AJHG* 2017; Fortes-Lima et al. *AJHG* 2017) and with simulations (Uren et al. *BMC Genetics* 2020), ancestry estimates that were estimated using ADMIXTURE and RFMix analysis generally correlate well. Uren et al also showed that RFMix outperforms ADMIXTURE with closer predictions of simulated data in their simulation studies. In the figure shown below, we compare our admixture fractions estimated using genotype-based ADMIXTURE and haplotype-based RFMix methods for all Bantu-speaking individuals. The correlation is generally very good, but in two cases RFMix assigns larger admixed ancestry fractions compared to ADMIXTURE: East-African and Khoe-San ancestries. BSP with these two ancestries might thus be overmasked. However, because our aim was to remove as much as possible non-Bantu admixture from BSP genomes, over-masking is a conservative scenario for our purpose. These plots were added in as new **Fig.S 4.4**. We also added a description at the end of the Materials and Methods section 1.9.

Fig.S 4.4 | Figure showing comparisons between admixture fractions in Bantu-speaking individuals estimated using ADMIXTURE and RFMix.

c. I could not find Figure S4.4 and S4.5 (cited on page 4 in the main text) in the supplement. The supplement goes from Fig.S 4.3 to Fig.S 5.1.

Thank you, we corrected the numbers accordingly. Those are **Fig.S 4.1** and **4.2**, respectively.

3. EEMS analysis:

a. Would an approach that uses identity-by-descent segments for use in EEMS like in Al-Asadi et al (2019, PMID: 30640906), be helpful for further analysis of the possible barriers to migration in the proposed interaction zones presented in Figure 5? This might help assess if the migration barriers estimated by EEMS were consistent throughout the past ~75-100 generations or changed over time.

We agree with the reviewer's comment. MAPS analyses (Al-Asadi et al. *PLoS Genet* 2019) could provide further insights into the observed migration rates. We now included MAPS results in **Fig.S 10.12**. We also added a description to the Materials and Methods section 1.17 and a discussion to the end of **Note.S 10**. There are similarities between the migration surfaces of EEMS results (**Fig.S 10.8**) and the new MAPS results for both the IBD length category defining older generations (2–4cM) and more recent generations (>6cM), around 56 and 13 generations ago respectively. Both these IBD categories indicate regions of lower gene flow overlapping with the geographic location of Zambia and higher gene flow along the coasts (although not completely overlapping). Interestingly the MAPS analyses of IBD capturing more recent generations show barriers in the DRC across the rainforest while in the analyses capturing older generations there seems to be increased gene flow across the rainforest. For populations toward the periphery of the BSP expansion it is unclear what the older generation MAPS analyses indicate as these populations (or part of their ancestry) might not all have reached their current geographic locations yet. Thus comparing time serial migration surfaces might make more sense for the areas around the BSP homeland.

EEMS analyses (left; Fig.S 10.8) compared to MAPS analyses capturing older generations- IBD tracts 2-4cM (center; Fig.S 10.12a), and to MAPS analyses capturing more recent generations - IBD tracts >6cM (right; Fig.S 10.12b).

b. Could the migration barrier inferred between the populations in South Africa/Mozambique and the population in Botswana (Figure 5c & S10.7c) be impacted by the sparse geographic sampling between these locations? I find this migration barrier interesting because Figure S10.7d shows that these populations have similar ADMIXTURE profiles. Could this also apply to why the barrier to migration is found in Zambia/DRC compared to the high migration zone found on the Indian Ocean coast where the geographic sampling density is higher?

Sample distribution might have an effect, however the EEMS method should correct for this since the aim of the method is to correct for sampling bias by estimating effective migration rates between populations while accounting for uneven sample sizes and geographic distribution. For the examples that the reviewer highlighted: In Zambia and DRC, we have higher resolution analyses of the region in the form of ADMIXTURE profiles that include more populations with small sample sizes (less than 10 samples per population). In **Fig.S 3.15**, the ADMIXTURE profiles were consistent with population

structure being present in these countries and different cluster contributions can be seen between populations from western and eastern regions in both countries, which aligns with the migration barriers inferred by EEMS. We agree with the reviewer that the Botswana (Ghanzi) population shows similar ADMIXTURE profiles to BSP in Mozambique and South Africa. This aligns with the fact that this population is likely Tswana (southeastern BSP). It might be the reason why the method indicated a blue “island” of high gene-flow around this population surrounded by areas of low gene-flow. It might be that the barrier indicated here is rather driven by the many southwestern BSP (bright blue dots in **Fig. 5c**) present in Namibia, Angola and western Zambia. Finer resolution with more populations from Botswana and Zimbabwe might help to refine inferences across this region in the future.

This explanation was added to the Supplementary discussion in Note.S 10. We thank the reviewer for highlighting this aspect.

4. On page 4 it is noted that the Herero and Himba show especially strong signals of founder effects compared to the rest of the Bantu speaking populations. It is reasoned that this could be due to “the consequence of genetic isolation since their arrival in southwestern Africa and recent endogamic practices linked to cattle herding.” Are there additional explanations or factors that could be impacting this? I am specifically thinking of the Herero genocide in the early 1900’s. Could this also be contributing to the signals of high ROH and high inbreeding coefficient? Or is the early 1900’s too early of a time frame for this to impact your results?

Since the populations from Namibia were sampled in the 1980s (and individuals were already adults), the sampled Herero individuals from this study would correspond to the first or second generation after the German genocide during the colonial period. After such a recent demographic bottleneck, we do not expect a signature of increased ROH, because the individuals remaining after the bottleneck still retain the diversity of the population before the bottleneck. This is one of the reasons why methods for demographic inference using the same kind of information (e.g. approaches based on sequential Markov coalescence (SMC), Li and Durbin *Nature* 2011) are not sensitive to recent population size changes. Therefore, we expect that the ROH patterns observed for the Herero population are due to older and potentially more prolonged processes. In addition, these ROH patterns are not specific to Herero only; similar ROH patterns are observed in other populations from Namibia such as Himba and Damara (**Fig.S 5.6b**, also included below), suggesting that common features, such as the pastoralist way of life, are more likely to be the main factors. A note commenting on this is now included in Supplementary Note.S 4 (last paragraph) as well as one sentence in the main text “*The bottleneck due to the early 20th century Herero genocide pursued by imperial Germany is not expected to trigger increased ROH as methods for demographic inference prove to be insensitive to recent population size changes (Note.S 4, Suppl. Material).*”

Fig.S 5.6b | All categories of ROH length for the Only-BSP dataset.

5. Figure S3.4 shows that K=16 has the lowest cross validation, yet Figure 2a presents the ADMIXTURE results for K=12. Why was K=12 chosen to display in the main text instead of K=16?

The reviewer is correct in that we chose to show K=12 instead of K=16 in the main text. As shown in Fig.S 3.4, K=12 also had a low CV and the additional structure that was inferred for K=13 to K=16 was not relevant for the genetic variation within Bantu-speaking populations, they included genetic structure within Nilo-Saharan speakers from Chad (teal component), Afro-Asiatic speakers from Ethiopia (green component), North and South Khoe-San speakers (rose-brown and pink components, respectively) (see Fig.S 3.9). We have added a sentence explaining this to the Supplementary methods (Section 1.7 on Unsupervised clustering analyses). We also now specified the population with the highest values for each component in the legend of Fig.S 3.8, so that it is clear which ancestry components are specified by K=12 and K=16 respectively.

Minor Comments

6. It looks like the setup of the F3 statistic is in the form of the out-group F3, where larger values indicate the two populations are more closely related. If you flipped the F3 in Figure S3.16 to be f3(Target; Yoruba, Amhara) would the values that are much greater than 0 in the current figure flip to being negative, therefore indicating that the Target has admixture between the Yoruba and Amhara?

We followed the reviewer's suggestion and calculated the f3-statistics in the form f3(Target; Yoruba, Amhara). We report below the original f3 tests, in the form f3(Yoruba; Amhara, Target) as in Fig.S 3.16 (left plot), and the new ones, in the form f3(Target; Yoruba, Amhara) (right plot). As expected, most positive values in Fig.S 3.16 (indicating the highest affinity of the Target with the Amhara, relative to Yoruba) change to negative ones when the test is performed in the form f3(Target; Yoruba, Amhara). As the reviewer points out, this indicates that the target has the highest affinity with both the Yoruba and Amhara (see figures below). These populations are Eastern BSP from Kenya, Rwanda and Uganda. Their Yoruba-Amhara mixed ancestry reflects the original WCA ancestry that they brought via the Bantu migration and Afro-Asiatic admixture, respectively, in

agreement with other genetic evidence we discuss in the main text. We have added the additional plot to **Fig.S 3.16** (panel c), and include a brief discussion of the significance of these findings in the figure legend.

7. Has the data upload process/application already been started with the European Genome-phenome Archive for the novel data?

The EGA submission process was started before submitting the manuscript for publication, and all the novel data presented in this study will be made available on EGA upon publication, for both modern-day and ancient DNA data.

Referee #3 (Remarks to the Author):

This paper reports a large genetic dataset of Bantu speaking populations (BSP) throughout Africa. It provides many descriptive analyses of how genetic similarities and diversity among BSP decline with time and geographic distance from their presumed west African origins. It provides some analyses of probable routes of migration as the Bantu swept out of West Africa around 5kya eventually ending up in present day South Africa.

These questions have been investigated by previous authors and this poses some difficulties for the current paper in identifying what is new about their contribution. Two papers using linguistic and geographical data (Grollemund et al 2015 and Koile et al 2022), both of which the authors cite, provide detailed analyses of the timings and routes of the Bantu migration, and the current paper does not add any new insights to these (a third previous paper – Currie et al, 2013 – does not provides dates but mostly agrees on the route of migration).

We appreciate the reviewer's acknowledgement of the significance of the Grollemund et al. (2015), Koile et al. (2022) and Currie et al. (2013) studies, which all explore the topic of the expansion of Bantu-speaking people. It is important to note, however, that those studies primarily focus on present-day linguistic data (this is now specifically indicated in the main text at the end of paragraph 1), whereas our study primarily analyzes genetic data, which are enriched by comparisons with linguistic inferences. By focusing on genetic data, our research introduces an additional dimension to the understanding of the expansion of BSP. This may or may not align, or only partially align, with linguistic inferences. Languages (and artifacts) can move without people moving, but genes cannot, thus genetics provides clear and unique insights into the movements of people. Therefore, we firmly advocate the separate and independent analysis of genetic and linguistic datasets, with comparisons conducted only after independent analyses have been performed. From a genetic standpoint, our study represents a substantial contribution, as it is currently the most extensive genetic investigation conducted to date on the expansion of Bantu-speaking populations.

Furthermore, the spread-over-spread events signaled in our genetic data (and shown additionally for archaeological data by Seidensticker et al. 2021) suggest that phylogeographic analyses based on lexical data from modern languages, such as those used by Grollemund et al. (2015) and Koile et al. (2022), are possibly not the best suited to reconstruct with exactitude the original migration routes of ancestral BSP, as we now point out more explicitly in the main text (see newly added text highlighted in section "Spread-over-spread events versus genetic continuity" in the main text).

In fact, the current paper suggests a date in the abstract for the origin of the Bantu expansion (the Bantu ingroup) that is in my view too young, and it is at odds with the two previous papers above, both of which find close agreement on this date. The authors do not provide any reason to prefer their suggested date over that of previous studies, and I might have missed it but I could not even find any direct evidence about dates in the paper or the 120 pages of SI. There were some materials on dates in Table S.7 but they are given in generations and I was not able to relate them to the trees they presented in the Supplementary Figures.

Thank you for noting this. We would like to clarify that the mentioned date in the abstract of approximately 4,000 years ago does not originate from the genetic data analyzed in our study. Instead, we included a widely utilized estimate based on existing archaeological and linguistic

evidence. In response to the reviewer's concern, we have revised the abstract accordingly, and the date now encompasses a broader range of 6,000–4,000 years ago to account for the uncertainty surrounding the expansion date. This range includes the linguistic date estimates mentioned in the two referenced papers.

Our study did not directly estimate an expansion date due to the nature of the analyzed SNP-array data, which is not suitable for precise expansion date estimates. However, we note that a genetic-based estimate of the expansion of BSP does exist. The study conducted by Li et al. (*Proc Biol Sci* 2014) utilized an Approximate Bayesian Computation (ABC) approach with microsatellite data, and their estimated expansion date was approximately 5,600 years ago. The broad range included in our revised abstract now also encompasses this genetic-based estimate.

We believe that our updated abstract provides a more accurate representation of the available evidence from both linguistic and genetic perspectives.

The same points could be made about the proposed migration routes. The previous papers broadly agree on routes while differing somewhat on one of the early timings. The current paper does not provide any grounds for preferring one of the previous works over the other, neither does it provide credible new routes.

Thank you for raising this point, and we appreciate the opportunity to further clarify the role and significance of our genetic analyses within the broader context of the expansion of Bantu speakers.

As mentioned in our previous response, linguistic and genetic data may reflect distinct underlying histories and might not necessarily align. Languages can move without people, and people may move and admix without influencing languages, so it is important to analyze languages and genes separately. However, we would like to emphasize that our study does provide genetic-based predictions of migration routes, which are illustrated in Figure 5.

In Figures 5a and 5b, we adopt a direct approach to trace the nearest F_{ST} genetic distance measures between populations, similar to the methodology employed before, e.g., Figure 2 in Grollemund et al. 2015, which was based on linguistic data. This approach allows us to infer potential migration routes based on genetic distances, reflecting the movements of people.

Additionally, in Figures 5c and 5d, we present alternative visual representations to indicate possible routes by depicting the genetic relatedness of the Bantu-only genetic component across geographic space.

These figures offer insights into the genetic aspects of population movement and serve as a complementary analysis to the linguistic data presented by Grollemund et al. and Koile et al. Consequently, our genetic analyses contribute to a more comprehensive understanding of the intricate dynamics surrounding the expansion of BSP, which was both a demic and cultural (linguistic) event and hence cannot be studied with one type of data only.

The current paper asks whether there was genetic interbreeding between BSP and non-BSP groups along the way and asserts that “We show for the first time that genetic diversity amongst Bantu-speaking populations declines with distance from western Africa”. This is misleading. Both previous genetic studies of the Bantu that the authors cite (authors' references 22 and 23) conclude that there has been substantial admixture between BSP and non-BSP people at a variety of sites around Africa. Indeed, this admixture is very well known among geneticists because it has confounded for

a long time attempts to study contemporary hunter-gatherer populations in southern Africa such as the San people, or among groups like the Sandawe or Hadza in East Africa.

We agree that our study is not the first to demonstrate admixture between Bantu-speaking populations (BSP) and non-BSP, and we do not claim this in our manuscript. Previous studies, such as References 22 and 23, along with several others, have reported on admixture events between BSP and non-BSP groups. However, our research uniquely highlights a decline in genetic diversity from West Africa when we specifically account for the admixture contribution that introduces additional diversity. By removing this admixture component, we observe a reduction in genetic diversity that has not been previously documented, which is what we refer to in the sentence “*We show for the first time that genetic diversity amongst Bantu-speaking populations declines with distance from western Africa*”.

This finding sheds new light on the complex interplay between admixture events and genetic diversity patterns within BSP, revealing the impact of admixture events on the overall genetic landscape. Our study, therefore, adds a valuable perspective to the existing body of research by uncovering this decline in genetic diversity when considering the influence of admixture in West African populations.

We hope this clarification provides a clearer understanding of the unique contribution of our study.

Now, admixture alone does not entail loss of diversity, but admixture does point towards relatively small migrant groups moving into occupied territories, rather than a huge invading force that swamped or drove the local inhabitants out. And the authors find a number of demographic founder events. Small groups, of course, lose diversity via sampling and so showing here that the Bantu genetic diversity declines with distance is a very minor addition, especially as the slope of the line is shallow, and this finding is not used to make points about social structure or perhaps adaptation.

We agree with the reviewer that it is theoretically expected for migrating populations to experience a loss of genetic diversity as they move away from their homeland. However, until now, this phenomenon has not been demonstrated for the expansion of BSP using genetic data, primarily due to the confounding effect of admixture with local groups and/or the insufficient geographic representation of previous studies.

In our study, we were able to address this challenge by effectively masking out the influence of admixture. The impact of admixture becomes evident when comparing the slope of the fitted line to genetic diversity as a function of distance from West Africa, before and after the removal of admixed fragments (**Fig. 4c, d**). We observed that, upon eliminating the influence of admixture, the slope of the line became steeper. This signal indicates a progressive loss of genetic diversity as Bantu-speaking populations migrated from West Africa during the expansion.

This “serial founder” signal has been previously documented for the out-of-Africa migration of modern humans, where genetic diversity declines with distance from East Africa, but its presence had not been previously detected in the context of the Bantu expansion. Our results also highlight the importance of considering the influence of admixture and its impact on genetic diversity patterns during population migrations.

The authors study the admixture further and conclude that the pattern of results that describes how Bantu populations’ similarity to their forebearers declines with time and space supports a serial founder effects model over a simpler isolation by distance model. I am prepared to believe that it might, but the authors do not distinguish this result from a simpler bottleneck model.

We appreciate the reviewer's comment and their consideration of alternative models.

The observed gradual decline of genetic diversity from West Africa to other regions in Africa provides support for a serial founder model, rather than a simple bottleneck model. This is further supported by the large geographic distances over which the Bantu-speaking populations expanded. A single bottleneck is unlikely given the substantial time and multiple generations it took for Bantu speakers to expand from their origin in West Africa to the rest of sub-equatorial Africa (as seen in independent data from the archeological record).

Additionally, the gradual decline in admixture dates with local groups, as we move further away from West Africa, reinforces the notion that a single bottleneck is unlikely over such extensive geographic distances. A simple or single bottleneck event would be more suitable for a scenario where a subgroup from a continental population colonizes a small island rather than the expansive migration of BSP. In this respect, the spread of BSP mirrors the spread of modern humans out of Africa, which involved humans spreading over substantial distances and involving multiple generations, and is best modeled by a serial bottleneck.

By considering these factors and the gradual decline of genetic diversity and admixture over distance, we find stronger genetic support for the serial founder model rather than a simpler bottleneck model. We appreciate the reviewer's suggestion, and we hope that this clarification better addresses the distinction between these models and their applicability to our findings. We added a paragraph to supplementary Note.S 7, which addresses this aspect.

Even if they could distinguish the two, this (serial founder effect) is a minor statement and the authors do not link it to anything important anthropologically or archaeologically. If people did pause, why, what were the implications, and what made them important. For example, the Grollemund et al paper finds empirical evidence that when Bantu peoples moved into rainforest, they did so more slowly than when they moved within savannah. The interest here is that Bantu were farmers, accustomed to savannah but not to rainforest. Moving into the rainforest then probably required the acquisition of new knowledge and skills.

We appreciate the reviewer's feedback and interest in the anthropological and archaeological implications of our findings. We agree that it is important to link the concept of serial founder effects to other factors such as the pace of migration and its implications. We do indeed consider the pace of migration (last paragraph in section "Routes and timing of expansions of Bantu-speaking populations"); the fact that overall there was a relatively constant rate of movement is rather remarkable, given the variety of environments that the Bantu-speaking people moved through.

It is also noteworthy that, in spite of this overall average constant rate of movement, we found evidence of older admixture in certain western regions, such as the admixture between Bantu-speaking populations and Western rainforest hunter-gatherers (RHG) in western DRC, while in certain eastern regions, the admixture appeared to be more recent, involving different eastern African groups in Uganda and Kenya. These findings suggest that the rates of movement into these specific regions may have been faster or slower than the average speed, or that admixture occurred earlier or later after the arrival of BSP compared to other regions.

We acknowledge the importance of considering the socio-cultural and environmental factors that may have influenced the movement of Bantu-speaking peoples into different regions of sub-equatorial Africa. Yet, in order to do that in a thorough manner we would need more in-depth archaeological and linguistic data (cf. Seidensticker et al. 2021, Pakendorf et al. 2017), which are currently unavailable for such an extensive coverage of Bantu-speaking populations. Thus, while we did not extensively discuss differences in social interactions and the acquisition of new knowledge

and skills during the Bantu expansion into different environments in this manuscript, we realize the potential significance of future research into these factors. Accordingly, we added the following sentence to the main text:

“Further investigation into the sociocultural aspects of the interactions with the linguistically and culturally diverse populations and the environmental challenges encountered by BSP during their expansion, particularly in adapting to diverse ecological zones and acquiring new subsistence practices, presents a promising avenue for future cross-disciplinary research.”

The presentation of results I'm afraid reads like a group of authors trying out the assembled toolkit of genetic analysis techniques and just reporting what they find. All of the figures are difficult to study (try, for instance, to distinguish among the colours in the legend of Figure 2), their implications are not well drawn, and the figure captions are not helpful. They have a bad habit of referring readers to supplementary figures and tables to aid understanding of their main text figures. But Figure captions need to stand alone. Figures 5a,b supposedly have four colours – I can only discern three – and these figures are not nearly as informative as comparable figures in Koile et al. or Grollemund et al.

Thank you for the feedback regarding the presentation of our results and figures. We apologize for any confusion caused and acknowledge the importance of contextualizing our analyses and providing clear and informative visuals to convey our findings. We have carefully considered the reviewer's comments and have made several improvements to address the concerns raised.

In our revised manuscript, we have made efforts to put the genetic analysis techniques in the context of the underlying models, provide clearer explanations of our methodology, and explicitly state the implications of our results to help readers (see highlighted text in the main manuscript).

We have also taken the reviewer's suggestions into account and made enhancements to the figure legends, providing additional information that aids in the interpretation of the figures and removing extensive references to supplementary material.

We acknowledge the reviewer's concerns about Figures 5a and 5b and the clarity of the colors used. We have made improvements to enhance the contrast between the red and brown colors, making them more distinguishable. This modification will address the previous ambiguity and ensure a clearer representation of the data.

Reviewer Reports on the First Revision:

Referees' comments:

Referee #1 (Remarks to the Author):

The authors have addressed all my comments and the manuscript is in my opinion publishable in its current revised form.

Referee #2 (Remarks to the Author):

Fortes-Lima et al thoroughly addressed my comments. I agree with their interpretation of the new analyses/figures, and I appreciate the additions to the text that provided more justification/clarification for why certain parameters were chosen. I also agree with their assessment of IBDNe with respect to how sample size and admixture might impact their results. I do not have any additional comments regarding interpretation or clarification of analyses/methods. I only have two minor comments:

1. I cannot access the github repository that stores the code. I get messages saying the page does not exist.
2. I think either the "of" or "for" needs to be deleted from the Fig.S 6.3 caption: "Estimated ancestry-specific effective population sizes (Ne) of for"

Referee #3 (Remarks to the Author):

I appreciate the authors' replies but they still face the difficulty that their genetic analyses don't add to the existing knowledge we have of the origin and precise routes of spread of the Bantu, including the timings of those events. In fact, these questions are investigated less rigorously here than in the previous linguistic papers.

It is true of course that genes can tell a different story to the languages, but here they don't clearly add a new dimension. We have known about admixture between BSP and locals for quite some time, and the evidence here about spread-over-spread events is inconclusive.

Author Rebuttals to First Revision:

Reviewer Comments & Author Rebuttals

Referees' comments are in black and Authors' responses are in blue font.

Referee #1 (Remarks to the Author):

The authors have addressed all my comments and the manuscript is in my opinion publishable in its current revised form.

We thank the referee for the review of our manuscript

Referee #2 (Remarks to the Author):

Fortes-Lima et al thoroughly addressed my comments. I agree with their interpretation of the new analyses/figures, and I appreciate the additions to the text that provided more justification/clarification for why certain parameters were chosen. I also agree with their assessment of IBDNe with respect to how sample size and admixture might impact their results. I do not have any additional comments regarding interpretation or clarification of analyses/methods. I only have two minor comments:

1. I cannot access the github repository that stores the code. I get messages saying the page does not exist.

We checked the link and made sure it is accessible now (see: [https://github.com/Schlebusch-lab/Expansion_of_BSP_peer-reviewed_article](https://github.com/Schlebusch-lab/Expansion_of_BSP_peer-reviewed_article)).

2. I think either the “of” or “for” needs to be deleted from the Fig.S 6.3 caption: “Estimated ancestry-specific effective population sizes (Ne) of for”

This typo was corrected in current Supplementary Fig. 51.

Referee #3 (Remarks to the Author):

I appreciate the authors' replies but they still face the difficulty that their genetic analyses don't add to the existing knowledge we have of the origin and precise routes of spread of the Bantu, including the timings of those events. In fact, these questions are investigated less rigorously here than in the previous linguistic papers.

It is true of course that genes can tell a different story to the languages, but here they don't clearly add a new dimension. We have known about admixture between BSP and locals for quite some time, and the evidence here about spread-over-spread events is inconclusive.

To address the reviewer's remarks the following text was added to the section 'Spread-over-spread events vs. continuity':

"This raises questions about the reliability of using only lexical and geographical data from modern Bantu languages for phylogeographic analyses to depict the ancestral BSP migration 18,19. Contact and admixture between incoming and previously settled Bantu-speaking groups could lead to genetic data reflecting a mixture of migration events, while linguistic data may represent only the latest spread event."

This text was added to the Conclusion section:

"While our genetic findings provide less precision compared to existing linguistic models 18,19, they caution against relying solely on modern language data for tracing BSP dispersion due to potential spread-over-spread events and genetic admixture between linguistically distantly related BSP. Our genetic findings highlight the need for a comprehensive interdisciplinary study into how the demographic history of BSP influenced their language evolution."